# Design of a unidirectional water valve in *Tillandsia*

Pascal S. Raux[1], Simon Gravelle [1] & Jacques Dumais [1]*

The bromeliad *Tillandsia landbeckii* thrives in the Atacama desert of Chile using the fog captured by specialized leaf trichomes to satisfy its water needs. However, it is still unclear how the trichome of *T. landbeckii* and other *Tillandsia* species is able to absorb fine water droplets during intermittent fog events while also preventing evaporation when the plant is exposed to the desert's hyperarid conditions. Here, we explain how a 5800-fold asymmetry in water conductance arises from a clever juxtaposition of a thick hygroscopic wall and a semipermeable membrane. While absorption is achieved by osmosis of liquid water, evaporation under dry external conditions shifts the liquid-gas interface forcing water to diffuse through the thick trichome wall in the vapor phase. We confirm this mechanism by fabricating artificial composite membranes mimicking the trichome structure. The reliance on intrinsic material properties instead of moving parts makes the trichome a promising basis for the development of microfluidics valves.

[1] Universidad Adolfo Ibáñez Facultad de Ingeniería y Ciencias, Viña del Mar, Chile. *email: jacques.dumais@uai.cl

embers of the genus *Tillandsia* are known as atmospheric bromeliads because of their near-exclusive reliance on the trichomes covering their leaves to acquire water and minerals from their environment (Fig. 1a–c, Supplementary Fig. 1)[1–3]. The *Tillandsia* trichome is made of a shield of dead cells, with unusually thick outer cell walls, sitting atop a dome cell (Fig. 1d–f, Supplementary Fig. 2a–c). The dome cell, in turn, connects the outer shield to the mesophyll of the leaf via a short stalk of two living cells. Moreover, the thick cuticle of the leaf forms a tube around the dome cell and the trichome stalk, thus guiding the surface water to the foot cells, where it is absorbed internally (Fig. 1e, f, Supplementary Fig. 2b, c).

The structural basis for the asymmetric conductance of the *Tillandsia* trichome was first explored by Mez[1]. According to his observations on the cosmopolitan *T. usneoides*, the central shield cells act as an impermeable plug that sinks into the space around the dome cell when the environment is dry, thus preventing the outward diffusion of water. When wet, the trichome swells, lifting it above the epidermis and thus opening a path for the inward flow of liquid water. A century later, these movements are still held as the main mechanism allowing the trichome to toggle between high and low water conductances[3]. Yet, the extensive overlapping of the trichomes' wings would seem to present a major impediment to the postulated coordinated movement of the trichomes (Fig. 1c, Supplementary Fig. 3). Moreover, Mez' mechanism fails to explain why, in the absence of fog, the trichome does not wick up liquid water from the leaf's mesophyll, thus maintaining its walls hydrated and highly conductive. Based on our experiments on *T. aeranthos* and *T. landbeckii*, we attribute the valve action to intrinsic properties of the trichome elements, in particular, those of the thick walls of the central shield cells and the plasma membrane of the foot cells.

## Results

**Water transport asymmetry in *Tillandsia*.** To quantify the water flux asymmetry of *Tillandsia* leaves, we reproduced in the laboratory the typical alternation of short fog events (a few hours) and long droughts (several days) experienced by these plants (see Methods). Leaves accumulated water rapidly even when presented with conditions favorable for water absorption for only 5% of the time (Fig. 2a), in agreement with measurements performed on other *Tillandsia* species[4]. Quantification of water exchanges in *T. aeranthos* reveals a large water flux asymmetry: $Q_{abs} = +240$ mg m$^{-2}$ min$^{-1}$ vs $Q_{eva} = -3.2$ mg m$^{-2}$ min$^{-1}$ (Fig. 2a) (similar values were obtained for *T. landbeckii*, Supplementary Fig. 2d, e). Taking into account the difference in potential driving the movement of water during absorption and evaporation ($\Delta\Psi_{abs} = +1.2$ MPa vs $\Delta\Psi_{eva} =$

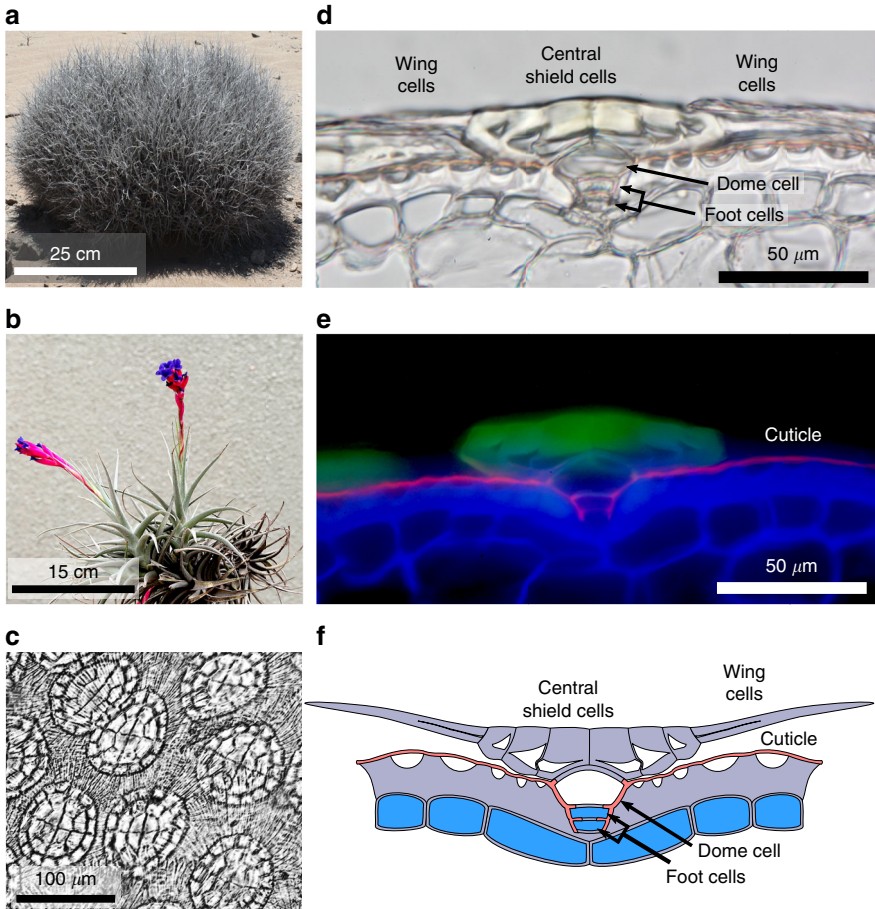

**Fig. 1 Structure of the *Tillandsia* trichome. a** *Tillandsia landbeckii* in the Atacama desert of Chile. **b** *Tillandsia aeranthos*, the plant studied most closely in this paper. **c** Surface view of the dense layer of peltate trichomes of *T. aeranthos*. Note the characteristic 4:8:16 organization of the central shield cells. **d** Transmitted light micrograph of a thin section of a *T. aeranthos* trichome. Note the unusually thick outer walls of the central shield cells, the thin wing cells, the dome cell, and the two foot cells connecting the trichome to the leaf mesophyll. **e** Composite fluorescence image of the trichome shown in **d**. The cell walls of the central shield cells and epidermis are highlighted by their own autofluorescence (green and blue) while the cuticle is marked with Sudan 3 (fuschia). The thin wing cells are invisible because of their minimal autofluorescence. **f** Diagram of the trichome structure. Living cells are shown in blue while dead cells have their lumen shown in white.

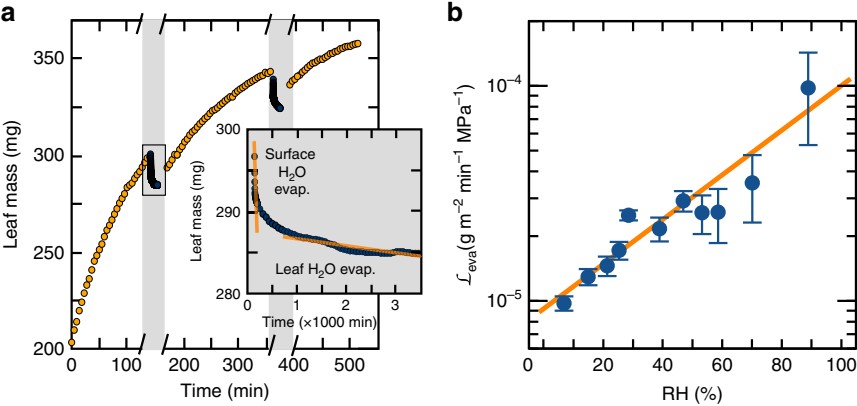

**Fig. 2 Water transport asymmetry in *Tillandsia aeranthos*. a** Water absorption and evaporation by a *T. aeranthos* leaf (dry weight = 61 mg, surface area = 10.4 cm²). The absorption time corresponds to the cumulative time spent immersed in water. Inset: zoom on the evaporation. After a short phase (less then 30 min) of quick evaporation of the free surface water, the mass loss by evaporation reaches a nearly constant rate corresponding the water loss from the leaf's mesophyll. **b** Overall conductance ($\mathcal{L}_{eva}$) of the *T. aeranthos* leaf as a function of the relative humidity (RH). The data points are shown as mean ± relative error ($n > 10$) and are fitted with an exponential function.

−92.8 MPa at RH = 50%) (Supplementary Note 1)[5,6]; we find a conductance for the absorption of fog water of $\mathcal{L}_{abs} = 200$ mg m⁻² min⁻¹ MPa⁻¹ and a conductance for the evaporation of water of $\mathcal{L}_{eva} = 0.034$ mg m⁻² min⁻¹ MPa⁻¹, where $\mathcal{L}_i = Q_i / \Delta\Psi_i$. These conductance values point to a remarkable asymmetry in overall leaf conductance of $\mathcal{L}_{abs}/\mathcal{L}_{eva} \approx 5800$. Furthermore, measurements of evaporative water fluxes at different relative humidities revealed another striking feature of the *Tillandsia* leaves - their ability to lower their water vapor conductance $\mathcal{L}_{eva}$ by a factor of 10 to face drier environments (Fig. 2b).

The conductances listed above are whole-leaf measurements and therefore hide potential contributions from the leaf's boundary layer and stomatal transpiration[7]. Stomatal transpiration acts in parallel with the evaporation at the level of the trichomes. We estimate that the conductance to water vapor of the trichomes would be as much as four times lower if the stomatal transpiration were removed (Supplementary Note 1). The resistance of the boundary layer acts in series with the resistance of the trichomes. Moreover, the boundary layer is present during evaporation but absent during absorption since liquid water is deposited directly on the leaf. Therefore, part of the conductance asymmetry measured could come from the boundary layer itself. The contribution of the boundary layer is easily shown to be of minor importance by measuring the water flux associated with the evaporation of the free surface water when the boundary layer is the only resistance to evaporation (Fig. 2a, inset; Supplementary Fig. 1). When acting alone, the boundary layer can sustain a water flux of $Q_{eva} = -2490$ mg m⁻² min⁻¹ while the characteristic flux for water evaporating from the leaf's mesophyll is only $Q_{eva} = -3.2$ mg m⁻² min⁻¹. We conclude that the conductance asymmetry reported above is imputable to the absorbing trichomes and must therefore be explained in terms of their structure.

**Mechanism for the trichome's conductance asymmetry.** The design of the trichome forces water to move along three successive structural elements before entering the mesophyll: the thick shield walls, the lumen of the dome cell, the semipermeable membrane of the outermost foot cell (Fig. 3a). When exposed to fog water, the highly hygroscopic shield walls absorb the droplets by capillarity, allowing a continuous path of liquid water from the surface of the trichome to the cytoplasm of the foot cells; the flow of water being driven inward by an osmotic gradient (Fig. 3b). When fog disappears and the effective water potential of the environment falls below the water potential of the foot cells, thousands of trichomes now

potentially act as capillary wicks (Fig. 3c). To avoid a wicking effect, it is crucial to rectify the movement of liquid water by making its outward flow impossible. Rectification requires to close two potential paths for the outward flow of liquid water. First, capillary flow within the wall space connecting the outer trichome to the mesophyll must be prevented. The *Tillandsia* trichomes achieve this with a precisely laid down cuticle covering the lateral walls of the dome cell and foot cells (Fig. 3d, Supplementary Fig. 2b; see also refs. [8,9]). Second, the liquid-gas interface must shift from the surface of the trichome shield to somewhere within the dome cell (Fig. 3c), so that the outward flow of water in the shield walls is by slow diffusion of the vapor phase instead of capillary wicking of liquid water. The liquid-gas interface will shift inwards if the rate of water evaporation at the trichome surface exceeds the rate at which water can be replenished by mass flow from the leaf mesophyll. To quantify the potential water flux through the trichome, we immersed leaves in $CaCl_2$ solutions of various osmotic potentials. We noted that the water flux across the trichome is linear with respect to the osmotic potential and changes sign at a critical concentration of 0.24 M, equivalent to −1.2 MPa of osmotic potential (Fig. 3e). Importantly, an external $CaCl_2$ concentration of at least 3 M (−14 MPa) is necessary to reach a liquid water flux equal to the rate of evaporation of free water at the trichome surface. Such a negative water potential cannot be generated in the liquid phase without using concentrations of solutes that far exceed the osmolarity of fog water. We conclude that the liquid mass flow through the trichome cannot match the typical rate of evaporation at the surface of the trichome. Thus, dry external conditions will necessarily force the liquid-gas interface to move deep into the trichome; ensuring that the outward movement of water involves diffusion of water vapor through the thick shield walls rather than capillary flow of liquid water.

To explain the role of the shield walls in resisting evaporation, we establish first that they are, as expected, highly hygroscopic (Fig. 3f, Supplementary Fig. 2f). The moisture content in the wall ($\mu$) is as much as 35% of the dry weight at RH = 100% but decreases sharply as the relative humidity drops from 100% to 50% (Fig. 3f, Supplementary Fig. 2f)[10]. Hygroscopic materials such as cellulose are known to show a water permeability varying with their own moisture content[11]. This singular physical property arises from the presence of hydroxyl groups on which water molecules adsorb, forming a third phase of bound water, alongside the vapor and liquid phases[12]. The presence of bound water swells the cellulose network thus increasing its porosity[13,14]. The net result of these interactions is a resistance to the

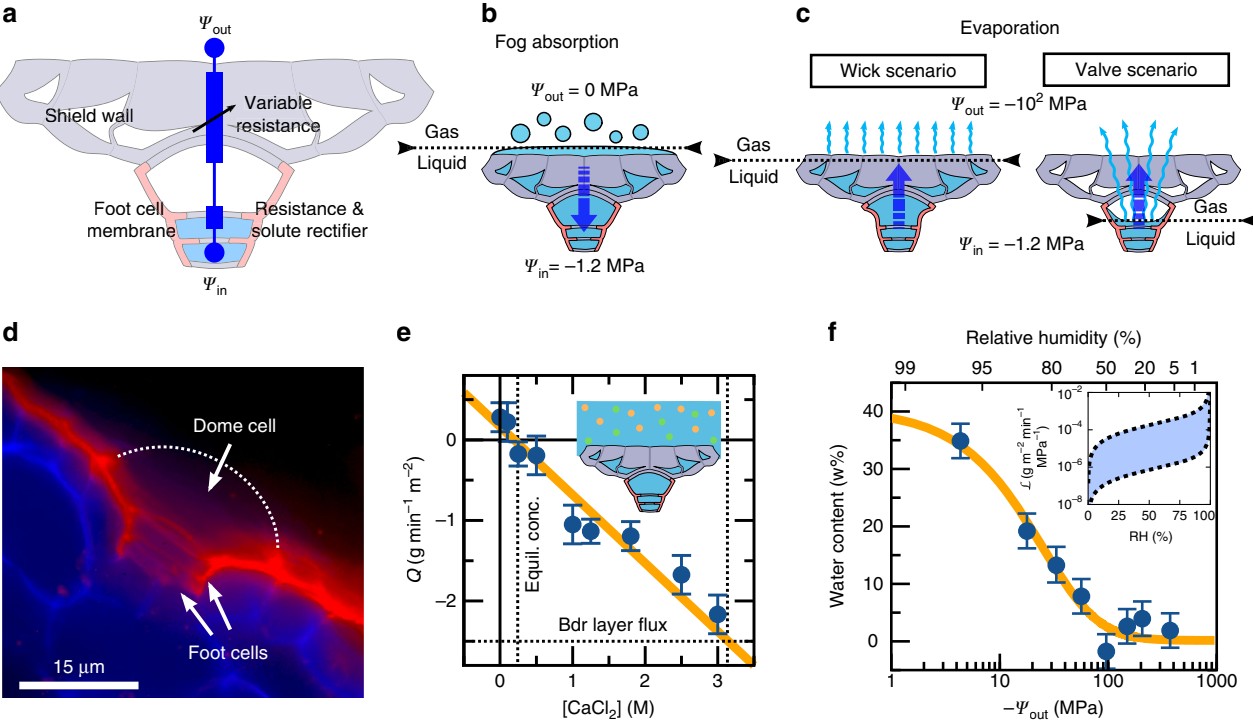

**Fig. 3 Conducting elements of the *Tillandsia* trichome valve. a** Equivalent electrical circuit for the *Tillandsia* trichome with the three elements contributing to flow asymmetry. **b** Flow of water during fog absorption. **c** Flow of water during evaporation under two distinct scenarios. **d** Thin cuticle layer lining the lateral walls of the dome cell and trichome stalk (red). The outer uncutinized wall of the dome cell is shown with a dotted line. **e** Water fluxes for *T. aeranthos* leaves exposed to $CaCl_2$ solutions (mean ± relative error, $n > 10$). At a critical concentration of 0.24 M (dotted line), the flow of water changes direction. In contrast, a concentration of 3 M is needed to maintain a liquid water flux equal to the boundary layer flux at the trichome surface. The slope is proportional to the plasma membrane conductance (see Supplementary Note 1). **f** Water sorption curve for isolated shield walls as a function of the atmospheric water potential and associated predicted cellulose conductance (mean ± relative error, $n > 10$). Inset: conductance as a function of relative humidity, see Supplementary Note 1.

transport of water that varies roughly exponentially with the moisture content of the cellulosic material (Fig. 3f inset, Supplementary Fig. 9). This variable conductance of the trichome shield could explain in part why the leaf's conductance to water vapor ($\mathcal{L}_{eva}$) depends strongly on the ambient relative humidity (Fig. 2b). In summary, our results support the following mechanism for the conductance asymmetry: the entry of water occurs in the liquid phase with the main resistance given by the plasma membrane of the foot cell (Fig. 3e and Supplementary Note 1)[15], while the exit of water takes place in the vapor phase with the main resistance to evaporation coming from the diffusion of water through the thick shield walls.

**Experimental manipulations confirm the transport mechanism**. Next we performed a series of experiments to confirm the functions of the three circuit elements of the trichome valve (Fig. 4a–c). According to the mechanism described above, the cellulosic shield offers little resistance to the absorption of liquid water but is the main barrier to evaporative losses. We tested this conclusion by carefully shaving leaves to remove the trichome shields (Supplementary Fig. 5). Results show that evaporative losses are increased by a factor of 560, while the rate of water absorption of the shaved leaves remains essentially unchanged ($P$ value > 0.2, Student $t$-test) (Fig. 4a, d). Another major feature of our mechanism is the role played by the plasma membranes of the foot cells in preventing the migration of solutes and liquid water from the mesophyll to the dome cell and shield. To test this assertion, we ruptured the membranes by freezing leaves at −80 °C, which keeps other structural features (cellulose shields, dome cell, cuticle) intact. This treatment causes a threefold

reduction in the absorption rate (Fig. 4d), in good agreement with the expected reflection coefficient of $\sigma = 0.3$ for the remaining cellulose cell walls[16]. More importantly, we measured a fourfold increase in the rate of water loss (Fig. 4b, d), which we explain by the spreading of osmotically active molecules from the foot cells to the dome cell and shield, leading to more capillary wicking of liquid water across the trichome. To confirm this conclusion, we exposed intact leaves to repeated immersion in a dilute solution of copper sulfate ($5 \cdot 10^{-3}$ M, $\Psi = -0.1$ MPa). The salt is expected to travel freely across walls but $Cu^{2+}$ should not cross membranes easily at this concentration[17]. Leaves were able to draw water from this weak $CuSO_4$ solution (Fig. 4c) but, as expected, $Cu^{2+}$ accumulated within the lumen of the dome cell rather than cross the cell membrane (Supplementary Fig. 6). The gradual accumulation of ions allows the dome cell and outer trichome to develop a more negative osmotic potential, thus maintaining liquid water at this level. The presence of liquid water explains the slight increase in the rate of evaporative loss after each exposure to the solution, reaching evaporation rates characteristic of the freeze-thaw experiments (Fig. 4c, d).

**Biomimetic device replicates the *Tillandsia* trichome valve.** Although the experiments reported in Fig. 4 demonstrate that the shield and plasma membrane of the foot cells are necessary for the proper functioning of the *Tillandsia* trichome, it is not clear that they are in themselves *sufficient* to achieve high asymmetry in water conductance. To confirm that the key elements have been identified, we built a macroscopic valve based on the design principles uncovered from the *Tillandsia* trichome. We assembled

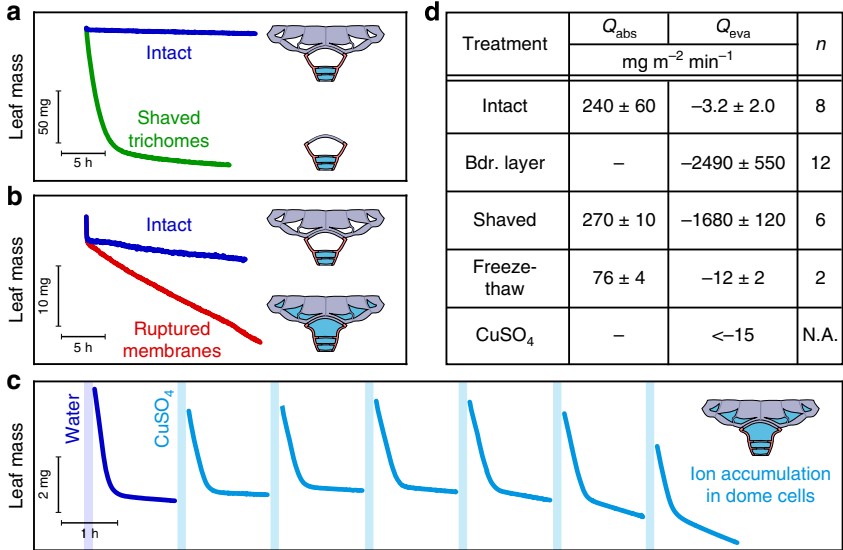

**Fig. 4 Response of the trichome valve to various experimental disruptions. a** Typical evaporation of a *T. aeranthos* leaf before and after shaving of the trichomes. The diagrams show the predicted state of the trichome during the evaporation phase. **b** Typical evaporation of a *T. aeranthos* leaf before and after freeze-thawing to rupture the plasma membranes of the foot cells. **c** Evaporation of a *T. aeranthos* leaf following successive 15-min immersions in a $CuSO_4$ solution ($5 \cdot 10^{-3}$ M). The first steep slopes correspond to the rate of evaporation of the free liquid film on the leaf surface immediately after immersion in liquid. **d** Summary table of the mean rate of absorption and evaporation under different treatments (mean ± std computed for a sample size of *n* leaves).

a composite membrane from a commercial semipermeable membrane and paper sheets, and used it to control water exchanges between the environment and a weak osmotic solution (Fig. 5a, Supplementary Fig. 7). As observed for the trichome, the presence of a cellulosic layer had almost no effect on the rate of water absorption by the system (Fig. 5b). However, when evaporating at a RH of 50%, the presence of a thick cellulose layer reduced the rate of evaporation by a factor of 14 (Fig. 5b, c), reflecting an asymmetry in the composite membrane conductance of at least 530. Although the asymmetry is still an order of magnitude lower than for *Tillandsia*, the difference is easily accounted for by the high permeability of the plasma membrane of cells, which is about one order of magnitude higher than the permeability of synthetic semipermeable membranes[18]. Finally, the conductance to water vapor decreased with decreasing relative humidity allowing the valve to reach its highest conductance asymmetry when exposed to the harshest external conditions (Fig. 5d). The biomimetic system thus reproduced the variable conductivity reported for the *Tillandsia* trichome (Fig. 2b).

## Discussion

The *Tillandsia* trichome illustrates the power of juxtaposing simple materials to achieve novel functionalities. Semipermeable membranes and cellulosic walls offer, on their own, no asymmetry to the transport of water. However, coupling of these structures results in a rectifiable composite membrane that can achieve a 5800-fold asymmetry in water conductance.

To evaluate the performance of the water capture mechanism evolved by *Tillandsia*, we compare it to the roots of desert succulents that are also known for their asymmetric transport of water[19]. The typical radial conductance of roots for the absorption of liquid water is $[6–15] \cdot 10^3$ mg m$^{-2}$ min$^{-1}$ MPa$^{-1}$ [20,21], which is at least 30 times higher than the whole-leaf conductance of 200 mg m$^{-2}$ min$^{-1}$ MPa$^{-1}$ measured in *Tillandsia*. However, while roots absorb over their entire surface, the trichomes' stalks represent only 2.6 of the total leaf surface area, the remaining 97.4 of the leaf surface being covered with an impermeable wax layer (Supplementary Fig. 4). Taking into account this reduction of the

surface area, the specific water flux for *Tillandsia* is $7.7 \cdot 10^3$ mg m$^{-2}$ min$^{-1}$ MPa$^{-1}$, which is well within the range reported in roots[20,21]. As such, the evolutionary innovation for the *Tillandsia* trichome does not reside in a higher conductance to absorb liquid water from the environment.

The true innovation emerges when considering how a low outward conductance is achieved. Although roots can show an outward conductance that is as much as 30,000 times lower than for the absorption conductance[19], this asymmetry is achieved over hours or days, and involves irreversible changes to the root cortex (formation of lacunae) and the stele (suberization)[22]. Moreover, recovery of a high absorption conductance is achieved only after regrowth of the roots[22]. Based on these observations, the water transport asymmetry observed in roots is more accurately described as a "hydraulic fuse" than a flow rectifier[23,24]. In contrast, the *Tillandsia* trichome is a true rectifier in the sense that it can toggle quickly and reversibly between high and low conductivity. Finally, we note that the rate of fog water interception by *Tillandsia* plants is comparable to meshes designed to capture fog water[25] (Supplementary Fig. 10) and that those rates of interception far exceed the rate at which water can be absorbed internally by the plant. It is therefore likely that *Tillandsia* satisfies its own water needs with fog water and also those of other plants and animals present in the vicinity.

## Methods

**Plant material.** Plants of *Tillandsia aeranthos* and *Tillandsia landbeckii* were kept indoors on a 13 h day:11 h night cycle and were watered with stagnant water twice weekly. The ambient temperature was maintained between 19 ℃ and 23 ℃, and the relative humidity ranged between 40% and 60%. Most experiments presented in this paper employ *T. aeranthos* because its large triangular leaves (1 × 5 cm) could be isolated individually and were easier to manipulate. However, some of the key experiments were also performed on *T. landbeckii*. In this case, small twigs comprised of several leaves were used.

Shortly before an experiment, leaves of *T. aeranthos*, about ∼5 cm in length, were carefully removed from the plant with a razor blade. The cut surface of each leaf was sealed with wax and a hook was embedded in the wax plug to allow weighing. After the end of each experiment, the leaves were cut along their length and kept for 24 h in an oven at 120 ℃ in order to measure their dry mass. The leaf area was calculated as $2(wl/2)$ where *w* and *l* represent the width and length of the triangular leaf shape. The prefactor 2 accounts for the two surfaces of the leaf.

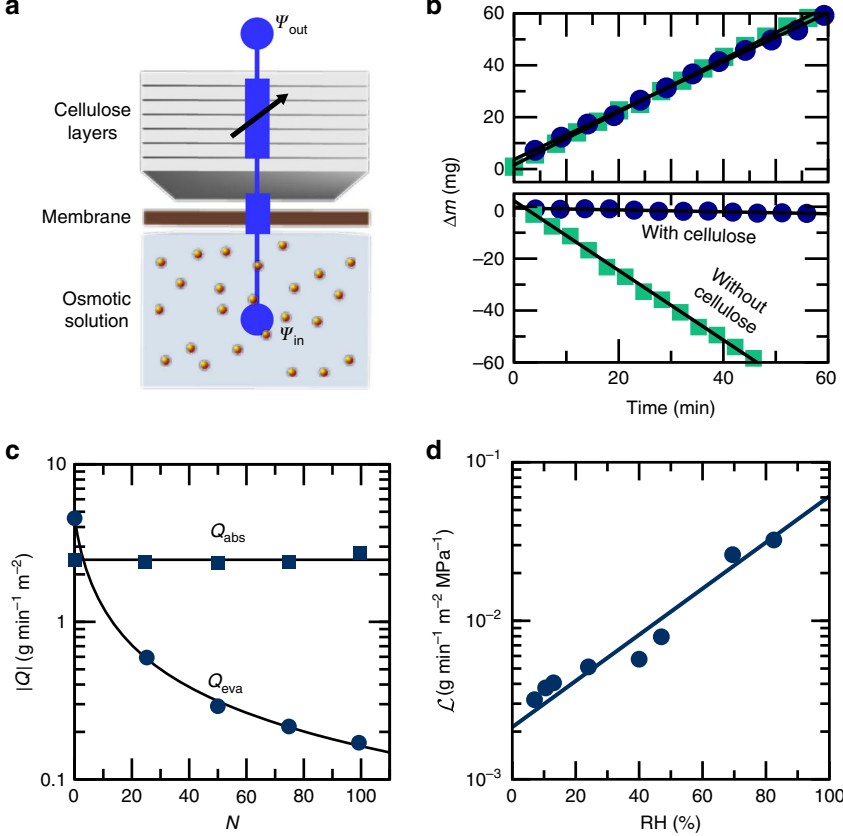

**Fig. 5 Water transport asymmetry in a biomimetic trichome. a** Diagram of the biomimetic system with the composite membrane made of a semipermeable membrane (Sterlitech SW30-HR) and the cellulose layer (paper sheets). A 1 M NaCl solution is enclosed behind this composite membrane. **b** Water absorption (top) and evaporation (bottom) of a biomimetic trichome in the presence (circle) and absence (square) of a cellulose layer ($\Delta m = m(t) - m(0)$) (RH = 50%). **c** Absolute value of the absorption (squares) and evaporation (circles, at RH = 50%) rates of the biomimetic composite membrane, as a function of the number $N$ of 100 $\mu$m-thick sheets of paper. **d** Effect of relative humidity on the effective water vapor conductance $\mathcal{L}$ of the biomimetic trichome prepared from 20 paper sheets.

**Microscopy**. In order to preserve the integrity of the cuticle layer for microscope observations, leaf samples were not fixed or embedded in paraffin; rather, fresh leaf samples were inserted between two close-fitting pieces of carrot and cut directly on a rotary microtome (Hacker Instruments Inc.). The leaf sections were stained for 3 min in a Sudan-3 solution (concentration 5 g/L in 70% ethanol) and then carefully rinsed with distilled water before being transferred to a glass slide with a brush. A 30% glycerol:water solution was used as mounting medium. Photographs were taken on an inverted Olympus IX-71 microscope with a Canon D200 camera. Fluorescence was observed under three different filters: TexasRed, FITC, and DAPI.

**Measurements of water uptake and evaporative water losses**. Water uptake experiments consisted of successive 10-min cycles, with each cycle starting with a 6-min immersion of the leaf in distilled water, followed by 2 min of centrifugation at 360 $g$ to drain the liquid film on the leaf surface. The remaining 2 min were devoted to weighing the leaf on an analytical balance (Shimadzu AUX120, Japan). A similar protocol was used for absorption data presented in Fig. 2 of the main text, except that the cycles were reduced to 5 min by shortening the immersion in water to 2 min and the manipulation time to 1 min. To measure the membrane permeability and estimate the osmotic potential of the leaf, some water uptake experiments were performed with immersion in CaCl$_2$ solutions of concentration ranging from 0 to 5 M.

Evaporative water losses were measured by weighing at 20 s intervals a leaf or several leaves suspended from the lower port of an analytical balance (Shimadzu AUX120) connected to a computer. Our experiments were conducted at a temperature of 22 ℃ ± 2 ℃. Before each measurement, the leaf was immersed in deionised water and centrifuged (see above) before weighting. While weighing, the leaves were kept within a custom-made chamber to control both the relative humidity and the boundary layer by forcing air circulation with a fan. Boundary layer contribution was estimated from the mass lost by evaporation just after immersion. To stabilize the ambient humidity and reach low relative humidities, we placed desiccant (silica gel beads from Dynamics Absorbent Inc.) at the bottom of the chamber. The level of hydration of the gel was changed to reach different relative humidities. For humidity higher than ambient (50–60%), we placed

distilled water at the bottom of the chamber. By controlling the surface area of the water bath relative to a small opening in the chamber, we were able to set any relative humidity intermediate between saturation and ambient. The temperature and humidity were recorded continuously with an Arduino equipped with a DHT22 sensor placed in close proximity to the leaves. Very slow variations of humidity were observed, typically 0.08% per hour, allowing us to measure evaporative water losses during extended periods of nearly constant relative humidity.

**Disruption of trichome functions**. Three series of experimental manipulations were undertaken to test the function of the different components of the trichome. In all cases, absorption and evaporation experiments were performed on the same leaf before and after the experimental treatment.

The first series of experiments involved freezing and subsequently thawing the leaves to rupture the plasma membranes without damaging other components of the trichome. For these experiments, a leaf was placed in a sealed plastic bag and kept for two hours at −80 ℃. Before absorption and evaporation measurements were performed, the leaf was allowed to equilibrate for one hour at ambient temperature while remaining in the sealed bag.

In a second series of experiments, the trichome shields on the abaxial face of the leaf were carefully removed with a razor blade while observing the leaf under a dissecting microscope. Care was taken to avoid damaging the cuticle. The integrity of the cuticle was checked in two ways. First, each leaf was kept for one day after performing the measurements, as preliminary tests had shown that damaged leaves display inhomogeneous shrinkage. When damage was evident, the data extracted from the leaf were discarded. Second, some leaves were stained with Sudan-3 and methylene blue dyes to confirm that the cuticle had not been damaged. Note that the flux reported for trichome-peeled leaves was calculated with the total surface of the leaf, whereas the trichomes were removed only on the abaxial surface of the leaf.

In a third series of experiments, we used a CuSO$_4$ solution to manipulate the osmotic potential of the dome cell. A leaf was first immersed in pure water and its rate of evaporative water loss was recorded for 90 min. Then, the leaf was

immersed in a solution containing $5 \cdot 10^{-3}$ M of CuSO$_4$ for 15 min, after which evaporative water losses were measured again. The last two steps were repeated several times to observed the cumulative effect of CuSO$_4$ exposure.

**Biomimetic asymmetric membrane**. The biomimetic system consisted of a tubular reservoir filled with ∼15 mL of a 1 M NaCl solution. One end of the tube was hermetically sealed while a composite membrane of area $\mathcal{A} = (3.8 \pm 0.3)$ cm$^2$ was glued at the other end of the tube using acetoxy silicone adhesive (Loctite SI 5398, Henkel, Germany). The composite membrane was made of a semipermeable membrane (SW30-HR, Sterlitech) covered with layers (20–100 sheets) of standard printer paper, where the thickness of each cellulosic layer is 100 $\mu$m. Printer paper is a natural choice for this preliminary biomimetic system because of its low cost and wide availability. Printer paper is made of a heterogeneous mixture of plant material, containing in particular cellulose, hemicelluloses, lignins, and some inorganic filling material such as clay or calcium carbonate[26].

For absorption experiments, the composite membrane was put into contact with a reservoir of distilled water; while for evaporation experiments, the membrane was exposed to a controlled atmosphere. In both cases, the mass $m$ of the reservoir containing the salt solution was recorded every 20 s on an analytical balance (Shimadzu AUX120) interfaced with a computer. The air in the chamber was stirred in order to ensure homogeneous humidity and temperature conditions within the chamber and to reduce the boundary layer above the membrane, and both temperature and relative humidity of the chamber were recorded continuously using an Arduino equipped with a DHT22 sensor.

**Reporting summary**. Further information on research design is available in the Nature Research Reporting Summary linked to this article.

## Data availability

The authors confirm that all data supporting the findings of this study are available within the manuscript and its supplementary files or are available from the corresponding author upon request. The data underlying all figures are available as a Source Data file.

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

## Acknowledgements

We thank Aymeric Pinel for help with the experiments and Robert Geoffrion for Fig. 1c and Supplementary Fig. 3. P.R. and S.G. acknowledge support from Fondecyt's post-doctoral fellowships #3150273 and #3170476, respectively. This research was supported in part by grant #1130129 from Fondecyt (Chile) and the US Army Research Development and Engineering Command (RDECOM), the US Office of Naval Research (ONR), and the US Air Force Office of Scientific Research (AFOSR) under grant number W911NF-16-1-0434 ($194,329USD).

## Author contributions

P.R. performed the experiments on *Tillandsia* and S.G. designed and tested the biomimetic system. P.R., S.G. and J.D. analysed the data and wrote the manuscript.

## Competing interests

The authors declare no competing interests.
