## [Peer Review File · Nature Communications]

Reviewers' comments:

Reviewer #1 (Remarks to the Author):

The manuscript describes the water absorption mechanism of 2 species of the genus *Tillandsia*. It demonstrates how the plants prevent evaporation via their absorption structures and finally the authors developed a biomimetic water absorbing trichome, based on the design principles of the *Tillandsia* trichomes. The findings are novel and will be of interest to a large field of readers. The manuscript is well written and findings and conclusions rest upon well elaborated experiments. However the reviewer recommends strongly to add the introduction and explanations, given in the supplemental materials section, to the main manuscript.

The reviewer has some further comments to improve and shorten the supplemental material

Main text and figures:

Missing line numbers on pages make it difficult to refer to the sentences!

Sizes of figures 1 A (insert) 1C and 1D are too small to see the details. Specially In fig. 1C the different layers mentioned in the legend should be marked in the figure. Were for examples are the "thin wing cells"? Represents 1C a light microscopy image of a section cut through a trichome? Are the data shown in fig. 1F based on own new data, or from other sources as indicated by the citation given in the text (but not in the figure legend)?

In fig. 1E the diagram of the trichome structure is composed of white, grey, red and blue colors. Blue represents living cells. Please explain why the epidermis cells with their cuticle on the outer side are not part of the living cells?

Fig 2 the reviewer recommends to separate A and D and increase the size of the remaining figures.

Pg. 7: please add the SI abbreviation for conductance after 530: However, when evaporating at a RH of 50%, the presence of a thick cellulose layer reduces the rate of evaporation by a factor 14 (Fig. 4B,C), leading to an asymmetry in membrane conductance of at least 530.

Materials and Methods

Pg. 11: Please give information about the cultivation conditions "Plants were kept indoors...."

Pg: 13: give number of layers of printer paper used and explain why printer paper has been used and not pure cellulose. Please give information about the chemical composition of the printer paper.

Supplemental materials

Pg 2: Chapter Liquid absorption: The authors say here "Specifically, we have shown that shaved leaves only increase their rate of water absorption by 12.5% as compared to intact leaves (Fig. 3D).", but state in the results on pg. 5 that the absorption was "essentially unchanged". 12.5% seems to be a remarkable change.

Figures S1 A and B need a scale bar and in S1 B the time for evaporation of the water film should be added under the single photographs.

Figure S2 please add a scale bar and give further information about the technique used to capture the photograph (light microscope or any other microscope?)

Figure S3 A and S3B staining of the trichomes is not mentioned. In B the mentioned structures are not visible. Both figures don not deliver new information compared to those already shown in Fig. 1 Please delete.

Figure S4 has a bad quality and bad resolution. Please remove it or change against a better one.

Fig. S5 The mentioned CuSO_4 crystals are not visible

Fig. S6: please explain in A what the grey layer under the trichome represents? If this layer represents the cuticle of the epidermal cells (in blue?) it's width is far too big (check your own data shown in S6 C. Please use one term in the legend and figure: here the cuticle thickness = cuticle width?

Reviewer #2 (Remarks to the Author):

The authors investigate water uptake in *Tillandsia* plants, which live in dry environments and rely on absorption of water from fog through trichome structures in their leaves, rather than getting their water supply from a root system. With a series of experiments, they determine that the trichomes act as a one-way valve allowing water to penetrate easily during the plant when fog is present, while preventing losses by evaporation when conditions are dry. The authors claim that this function is achieved thanks to a combination of several layers of different properties: a highly permeable cellulose membrane on the outside, and a semipermeable membrane on the inside, both being separated by an air space (lumen). During filling, liquid water permeates the whole structure and is sucked into the plant by osmosis, while when evaporation takes place, cavitation empties the lumen and the evaporation surface recedes to a small area within the structure, making water loss very slow. The authors finally propose a biomimetic structure made of cellulose layers (paper) and an osmotic membrane separated by an air space, which displays a strong asymmetry between absorption and evaporation of water.

The paper is well written, the experiments are carefully designed and executed (including repeated experiments with different samples), and the results should be of interest for different communities in biology, physics and engineering. Although most of the claims are convincing, I am uneasy with some of the conclusions, which might require clarification. I explain some of my concerns below.

One of the key ingredients, according to the authors, is cavitation of the water contained in the lumen. I don't see why this is necessary: couldn't the liquid-vapor interface recede continuously from the outside of the shield to the lumen through the cellulose layer when evaporation takes place? Of course the end result is the same, i.e. the liquid-vapor interface ends up at the location of the plasma membrane.

The second key ingredient put forward by the authors is the fact that the plasma membrane has a much smaller surface area, lowering evaporation compared to the case where it would occur on the outer surface of the trichome. While I agree that this effect can play a role, it seems to me that the main factor explaining the slow water loss is the fact that water can only exit in the vapor phase (by diffusion in the air in the lumen, then across the cellulose network) rather than by liquid flow during water intake. If this is true, it is not even clear if the lumen space plays that much of a role in the asymmetry of fluxes between intake and evaporation: diffusion of water vapor through the cellulose network could be the limiting step in transport whether there is an air space below or not. In this case, the difference between liquid water transport and vapor water transport in the cellulose wall could be the only "flow rectifier" component of the system, contrary to the claim that it is the lumen space that acts as a rectifier. Clarification on these aspects would be helpful.

Also, the simple biomimetic system proposed by the authors seems to be missing both key ingredients discussed above. It is not clear whether cavitation occurs in this system (is it possible to see the air space filling and emptying during operation?), and if I understand correctly, there is no difference in evaporation area between the cellulose layer and the semipermeable membrane. To demonstrate the role of the air space as a flow rectifier, the biomimetic experiments should also include a case where the air space is removed between the cellulose layers and the semipermeable membrane.

Another thing that was unclear is where the water absorbed during intake events is stored, both in real leaves and in the biomimetic system: is the osmotic solution diluted, and if so shouldn't the driving force decrease over time?

Finally, the authors discuss potential applications for water harvesting in the conclusion. A variety of designs have been proposed in recent years for fog harvesting and it would be useful to compare the strategy elucidated here to these other strategies. An obvious downside seems to be that since osmosis is the main driving force, the harvested water contains solutes that have to be filtered out later.

I also have a few more technical questions or remarks:

- What is the basis for defining water conductance with respect to water potential? It seems to me that this definition artificially amplifies the asymmetry between uptake and evaporation (the 5800 factor) due to the nonlinearity of water potential with RH.
- Is there any reason why (and could you confirm that) freeze-thaw procedures only rupture plasma membranes and not other components?
- The expected effect of CuSO₄ treatment should be described more clearly. Why would it increase evaporation flux? Why would it form crystals as in Fig S5? And why the choice of this particular compound?
- How are the water sorption measurements made? It's also not obvious how these results help understanding the function of trichomes, as they are not used in the discussion except for illustrating that the cellulose layers are hygroscopic.
- I did not understand the argument used by the authors against the "plug" scenario for the trichome function: overlapping of the trichome wings is not a problem if all the trichomes have a coordinated motion. Also, are the wings the only external extensions of the trichomes or is there a "hair" part not represented on the schematics?
- The methods or supplementary material would benefit from addition of a picture of the biomimetic device. Also, figures S1-S5 lack information of scale, and it's not obvious what we should be observing in the sequence in Figure S1B.
- Why would the pathways for evaporation (e.g. Fig. S7) be different from the pathways for water absorption?
- The authors mention concerns about the boundary layer outside of the leaf but then seem to exclude that contribution. Could you clarify whether the boundary layer is a concern or not and how it affected early measurements?
- Is the swelling of the cellulose network due to water absorption similar to the Bingham effect for porous media?

Reviewer #3 (Remarks to the Author):

In this work, the authors have argued that the asymmetric water transport found on *Tillandsia aeranthos* can be explained by the three distinct cell structures. Based on this understanding of the water transport under different conditions, the authors have further demonstrated that a synthetic unidirectional water transport valve structure in a larger length scale works. This work contains an interesting understanding of asymmetric water transport through *T. aeranthos* leaves but I am not sure whether the understanding is solid using the current set of experiments and information about them. I postpone my decision until the authors address the following additional comments and questions below. With the authors' revised manuscript, I will check the novelty, scientific approaches, and impacts of this work holistically.

1. I cannot find the temperature condition of experiments including Figures 1F and 1G. The authors should add the temperature conditions for all their experiments that are influenced by relative humidity. It would be also great if the authors can compare them with the actual temperature conditions in the Atacama desert of Chile.
2. Figure 4A should show the dimension of the composite membrane system, including the spacing between the cellulose layers and the bottom membrane. Showing the actual experimental setup is usually the best way. Figure 4B should show the relative humidity and other experimental conditions.
3. I am not sure whether the authors actually used *T. landbeckii* for their quantitative experiments because all the figures regarding quantitative information indicated only *T. aeranthos*. Please clarify this point.
4. One of the main factors regarding the evaporation suppression mechanism is the vacant space in the dome cell. If the authors's explanation is correct, it would be possible to visualize this space with and without water using other types of visualization technique such as 3D confocal microscopy. This would clearly support the authors' argument and significantly help readers understand the authors' explanation.
5. Figure 1A and S1-S5 should have scale bars. In particular, Figure S1B should show the time information on each image.

Design of a Unidirectional Water Valve in *Tillandsia*

Pascal S. Raux, Simon Gravelle, Jacques Dumais

Response to Reviewers

We thank the reviewers for their numerous and insightful comments. We have made many important changes to the manuscript in responding to these comments. To facilitate the review process, all significant changes to the main text were highlighted in blue.

Reviewer #1

The manuscript describes the water absorption mechanism of 2 species of the genus *Tillandsia*. It demonstrates how the plants prevent evaporation via their absorption structures and finally the authors developed a biomimetic water absorbing trichome, based on the design principles of the *Tillandsia* trichomes. The findings are novel and will be of interest to a large field of readers. The manuscript is well written and findings and conclusions rest upon well elaborated experiments. However the reviewer recommends strongly to add the introduction and explanations, given in the supplemental materials section, to the main manuscript.

We thank the reviewer for his/her positive comments about the paper. Many details that were formerly given in the supplemental materials are now part of the main text. In particular, we added:

- the measured leaf conductance for absorption and evaporation (lines 63-65)
- a more complete explanation of why the conductance asymmetry for whole-leaf measurements is also a conservative estimate of the conductance asymmetry for the trichome (lines 72-90)
- the calculation for the specific absorption conductance for the trichome (lines 214-227)
- the Methods (lines 250-364)

However, we have left many introductory sections in the Supplementary Note since they simply restate information already available in textbooks. We are open to include more information from the Supplementary Note into the main text if some specific sections can be pointed to. In its current form, the Supplementary Note extends over six pages and cannot all fit in the main text.

The reviewer has some further comments to improve and shorten the supplemental material

Main text and figures:

Missing line numbers on pages make it difficult to refer to the sentences!

Line numbers were added.

Sizes of figures 1 A (insert) 1C and 1D are too small to see the details. Specially In fig. 1C the different layers mentioned in the legend should be marked in the figure. Were for examples are the “thin wing cells”? Represents 1C a light microscopy image of a section cut through a trichome?

Fig. 1 was divided into two figures to allow more space to show clearly the trichome structure. The light micrograph of the trichome (Fig. 1d) is now labelled in the same way as the diagram (Fig. 1f), including the “thin wing cells” mentioned by the reviewer. Figure 1C (now Fig. 1d) is indeed a transmitted light micrograph of a thin section through the trichome. This was added to the figure legend.

Are the data shown in fig. 1F based on own new data, or from other sources as indicated by the citation given in the text (but not in the figure legend)?

The data shown in Fig 1F (now Fig. 2a) are our own. The previous sentence in the main text was confusing. We replaced it with the following sentence: “Leaves accumulated water rapidly even when presented with conditions favorable for water absorption for only 5% of the time (Fig. 2a), in agreement with measurements performed on other *Tillandsia* species [Bieb1964].”

In fig. 1E the diagram of the trichome structure is composed of white, grey, red and blue colors. Blue represents living cells. Please explain why the epidermis cells with their cuticle on the outer side are not part of the living cells?

The reviewer’s interpretation is correct, the lenticular spaces just below the cuticle (and right above the thick wall layer) are the epidermal cells of the leaf. The cells were depicted as “dead” because they show no cytoplasmic content in light microscopy. This conclusion was confirmed by staining thick leaf sections with FM1-43, a membrane dye. While the membrane of cells in the mesophyll is brightly stained, there is no evidence of staining in these epidermal cells.

Fig 2 the reviewer recommends to separate A and D and increase the size of the remaining figures.

Is the reviewer suggesting to make three separate figures? For the moment, we have left the figure (new Fig. 3) as it was because the sequence in which each panel is shown fits the logic of the presentation and all the panels of the figure pertain to the same question. If the figure needs to be divided, the panel sequence would have to remain intact.

Pg. 7: please add the SI abbreviation for conductance after 530: However, when evaporating at a RH of 50%, the presence of a thick cellulose layer reduces the rate of evaporation by a factor 14 (Fig. 4B,C), leading to an asymmetry in membrane conductance of at least 530.

Here we are referring to the ASYMMETRY in conductance (ratio of the absorption and evaporation conductances), thus the number is without dimensions.

Materials and Methods

Pg. 11: Please give information about the cultivation conditions “Plants were kept indoors.....”

We added the following information about the temperature and relative humidity (line 251):
“Plants of *Tillandsia aeranthos* and *Tillandsia landbeckii* were kept indoors on a 13h day : 11h night cycle and were watered with stagnant water twice weekly. The ambient temperature was maintained between 19°C and 23°C, and the relative humidity ranged between 40% and 60%.”

Pg: 13: give number of layers of printer paper used and explain why printer paper has been used and not pure cellulose. Please give information about the chemical composition of the printer paper.

The number of layers was varied during the experiments between 20 and 100. We added this information to the methods as well as the following sentences (line 349):

“Printer paper is a natural choice for this preliminary biomimetic system because of its low cost and wide availability. Printer paper is made of a heterogeneous mixture of plant material, containing in particular cellulose, hemicelluloses, lignins, and some inorganic filling material such as clay or calcium carbonate.”

Note that we have since obtained similar results using membranes made of pure bacterial cellulose.

Supplemental materials

Pg 2: Chapter Liquid absorption: The authors say here “Specifically, we have shown that shaved leaves only increase their rate of water absorption by 12.5% as compared to intact leaves (Fig. 3D).”, but state in the results on pg. 5 that the absorption was “essentially unchanged”. 12.5% seems to be a remarkable change.

Our characterization is based on the recorded means and standard deviations for the absorption of water by intact and shaved leaves. According to the table in Fig. 4d (formally Fig. 3D), the typical rate of water absorption by an intact leaf is $240 \pm 60 \text{ mg m}^{-2} \text{ min}^{-1}$ (n=8) while for a shaved leaf it is $270 \pm 10 \text{ mg m}^{-2} \text{ min}^{-1}$ (n=6). Using a t-test to compare the means, the P value is found to exceed 0.2. Therefore, the two means cannot be said to be statistically different. In contrast, the rate of evaporation is multiplied by a factor of 560 in the shaved leaves. We added the P value to the sentence in the main text to clarify on what ground we qualified the rate of absorption as “essentially unchanged” (see line 161).

Figures S1 A and B need a scale bar and in S1 B the time for evaporation of the water film should be added under the single photographs.

Scale bars were added and the time interval for the pictures in Supplementary Fig. 1b is now stated in the figure legend.

Figure S2 please add a scale bar and give further information about the technique used to capture the photograph (light microscope or any other microscope?)

This figure is now Supplementary Fig. 3. The scale bar was added. The picture was captured using standard transmitted light. This information was added to the figure legend.

Figure S3 A and S3B staining of the trichomes is not mentioned. In B the mentioned structures are not visible. Both figures don not deliver new information compared to those already shown in Fig. 1 Please delete.

This figure was added as a panel to Supplementary Fig. 4. The sample was stained with toluidine blue. This information is now indicated in the figure legend. Although it is true that the figure is not easy to interpret, we believe it serves an important function by showing, side-by-side the respective areas at various levels in the trichome path. Since the main function of these pictures is to allow the precise measurements of areas, it was merged with Supplementary Fig. 4, where the diagram of the cross-section of the trichome should help the interpretation of the pictures.

Figure S4 has a bad quality and bad resolution. Please remove it or change against a better one.

The transmitted light micrograph was changed for micrographs obtained with epifluorescence which, we believe, provide a much better view of the pegs left after shaving the trichomes of the leaf. The new micrographs now appear in Supplementary Figure 5.

Fig. S5 The mentioned CuSO_4 crystals are not visible

The CuSO_4 crystals, as such, are not visible so the word “crystals” was removed. We were referring to the accumulation of CuSO_4 just below the central shield cells. We have now put an untreated and a treated trichome side-by-side to highlight better the difference. These micrographs are shown in the new Supplementary Figure 6.

Fig. S6: please explain in A what the grey layer under the trichome represents? If this layer represents the cuticle of the epidermal cells (in blue?) its width is far too big (check your own data shown in S6 C. Please use one term in the legend and figure: here the cuticle thickness = cuticle width?

Fig. S6A is now Supplementary Fig. 4a. The large grey layer does not correspond to the cuticle but instead to cellulose walls. See Fig. 1d-f for guidance. The colors of the diagram were modified (with the waxy cuticle in red) to match the diagram in the main text and clarify the information. Also, we replaced the term “cuticle width” by “cuticle thickness” everywhere.

Reviewer #2

The authors investigate water uptake in Tillandsia plants, which live in dry environments and rely on absorption of water from fog through trichome structures in their leaves, rather than getting their water supply from a root system. With a series of experiments, they determine that the trichomes act as a one-way valve allowing water to penetrate easily during the plant when fog is present, while preventing losses by evaporation when conditions are dry. The authors claim that this function is achieved thanks to a combination of several layers of different properties: a highly permeable cellulose membrane on the outside, and a semipermeable membrane on the inside, both being separated by an air space (lumen). During filling, liquid water permeates the whole structure and is sucked into the plant by osmosis, while when evaporation takes place, cavitation empties the lumen and the evaporation surface recedes to a small area within the structure, making water loss very slow. The authors finally propose a biomimetic structure made of cellulose layers (paper) and an osmotic membrane separated by an air space, which displays a strong asymmetry between absorption and evaporation of water.

The paper is well written, the experiments are carefully designed and executed (including repeated experiments with different samples), and the results should be of interest for different communities in biology, physics and engineering. Although most of the claims are convincing, I am uneasy with some of the conclusions, which might require clarification. I explain some of my concerns below.

We thank the reviewer for his/her positive comments about our paper. We have tried to clarify the conclusions that were deemed poorly supported.

One of the key ingredients, according to the authors, is cavitation of the water contained in the lumen. I don't see why this is necessary: couldn't the liquid-vapor interface recede continuously from the outside of the shield to the lumen through the cellulose layer when evaporation takes place? Of course the end result is the same, i.e. the liquid-vapor interface ends up at the location of the plasma membrane.

We agree with the referee and believe the slightly different perspectives on this issue might be reconciled by agreeing first on a definition of "cavitation". The cavitation we postulate is commonly referred to as air-seeding (see: Tyree, M. T., & Sperry, J. S. 1989. Annual Review of Plant Biology, 40: 19-36, pages 27-29); that is, when the water tension in the lumen of the dome cell is sufficiently negative to "pull in" the weakest meniscus in the wall. We did not mean to say that cavitation would be of the homogeneous type. In fact, it is probable that the tension needed to pull the meniscus into the lumen of the dome cell is not very negative. For example, if the largest pores in the wall have a diameter of $d = 0.1 \mu\text{m}$, the Laplace pressure would be $P = -70[\text{N/m}] / d = -0.7 \text{ MPa}$. As such, if cavitation by air-seeding does occur, we do not expect the event to be very dramatic. On the other hand, it is not clear to us that the liquid-gas interface can simply recede continuously since the thick shield wall is a wetting material with low contact angle and therefore water tension is needed to pull the water menisci through the wall.

The reasons we believed that the water tension in the dome cell is not trivial and may deserve the term “cavitation” (rather than migration of the liquid-gas interface) comes from the structure of the dome cell itself. Its dome shape and solid anchor in the outer wall of the central shield cells suggest that it is designed to sustain a certain tension without collapsing (see Fig. 1d-f). Also, we would like to refer the referee to our papers on the mechanism of spore discharged in the fern leptosporangium (Noblin *et al.* 2012. The fern sporangium: a unique catapult. *Science* 335: 1322; Llorens *et al.* 2016. The fern cavitation catapult: mechanism and design principles. *Journal of the Royal Society Interface* 13: 20150930). The cells of the leptosporangium’s annulus undergo a transition very similar to the transition we postulated for the dome cell of *Tillandsia*. In the case of the leptosporangium, we were able to demonstrate that the menisci in the thin outer walls of the annulus are able to support a negative pressure of -10MPa before the whole cell cavitates; and do so repeatedly. Therefore, it is not remarkable to us that plant cells other than vascular cells could cavitate as a normal part of their function.

Nonetheless, as stated by the reviewer, the end result is the same whether the dome cell cavitates by air-seeding or not. Therefore, we modified the text to (1) explain better how we envision the movement of the liquid-gas interface during drying of the trichome and (2) leave open the possibility that the liquid-gas interface simply migrates “uneventfully” into the lumen of the dome cell. Specifically, we do not use the word “cavitation” directly:

(line 108). “Second, the water column passing through the lumen of the dome cell must be ruptured once the external source of water is lost. Liquid mass flow across the trichome is reversible, as shown by the water flux in or out of the leaf when the latter is exposed to CaCl₂ solutions of various osmotic potentials (Fig. 3e). Reversal of the flow occurs at a critical concentration of 0.24M, equivalent to -1.2MPa of osmotic potential. Wicking will occur if the dome cell can develop this water potential while filled with fog water. Since fog water has a minimal osmotic potential, a water potential of -1.2MPa would have to be achieved by a lowering of the hydrostatic pressure within the lumen of the dome cell. Although vascular elements routinely support tensions more negative than -1.2MPa [Tyree1989], such tension within the un lignified walls of the dome cell is likely to result in air seeding, that is, the movement of a water-air meniscus within the lumen of the dome cell. In that context, it is noteworthy that the lumens of the shield cells and dome cell form a conduit through which repeated air-seeding events across thin walls would bring the liquid-gas interface to the surface of the outermost foot cell (Fig. 1d,f; Supplementary Fig. 2a,c). The inward migration of the liquid-gas interface reduces the effective surface for evaporation to less than 3% of the total leaf surface area (Fig.~3d, Supplementary Fig.~4) and adds the thick shield walls as a new resistance for the outward diffusion of water vapor (Fig. 3a,c). These two effects explain the importance of rectifying the flow of liquid water at the level of the dome cell for the trichome to act as a valve.”

The second key ingredient put forward by the authors is the fact that the plasma membrane has a much smaller surface area, lowering evaporation compared to the case where it would occur on the outer surface of the trichome. While I agree that this effect can play a role, it seems to me

that the main factor explaining the slow water loss is the fact that water can only exit in the vapor phase (by diffusion in the air in the lumen, then across the cellulose network) rather than by liquid flow during water intake. If this is true, it is not even clear if the lumen space plays that much of a role in the asymmetry of fluxes between intake and evaporation: diffusion of water vapor through the cellulose network could be the limiting step in transport whether there is an air space below or not. In this case, the difference between liquid water transport and vapor water transport in the cellulose wall could be the only “flow rectifier” component of the system, contrary to the claim that it is the lumen space that acts as a rectifier. Clarification on these aspects would be helpful.

We agree with the core of the reviewer’s comment. The reduction in the surface area is not the main contribution to the transport asymmetry because under normal conditions the area of absorption and evaporation are the same and correspond to the area of the outermost foot cell (see below). Rectification is necessary, in part, to avoid the evaporation area to exceed the absorption area because of capillary wicking to the shield cells. The largest contribution to the conductance asymmetry comes from the thick shield walls which can let liquid water flow easily but presents a major obstacle for the diffusion of water vapor.

Having said that, we would like to clarify our choice of words and make one more substantive point. We use the expression “rectifier” in the same spirit as Park Nobel (Nobel, P. S. & Sanderson, J. Rectifier-like activities of roots of two desert succulents. *Journal of Experimental Botany* 35, 727–737, 1984) and, we believe, in accordance with the standard electrical meaning. According to the Oxford dictionary, a rectifier is:

“An electrical device which converts an alternating current into a direct one by allowing a current to flow through it in one direction only.”

A rectifier does not contribute to the resistance of a flow path and only dictates in what direction the flow can occur. In the trichome, the dome cell appears to serve the role of a liquid water rectifier while the shield walls play the role of a variable resistance. This main conclusion is captured in the electrical diagram of Fig. 3a (former Fig. 2A).

The point of contention, we believe, is whether rectification by the dome cell is, in and of itself, a noteworthy feature of the *Tillandsia* trichome. To put it differently, one could ask whether the wick scenario illustrated in Fig. 3c is in fact a likely outcome if the trichome is poorly designed. We list below the reasons why we believe that rectification and a variable resistance are two distinct functions served by two distinct structures in the trichome.

First, the wick scenario does exist in plants. It is how water is lifted in trees. Therefore, it is not obvious that the gas-liquid interface in the shield wall must necessarily recede when the trichome is exposed to non-saturated air. The interface could simply stay put in the shield wall and pull water from the plant to replace water that has evaporated. If this were to happen, the conductance asymmetry would be lost.

Second, two of our experimental manipulations (freezing and immersion in CuSO₄ solutions) were attempts to disrupt the rectifying function of the dome cell while leaving the shield walls intact. In both cases, we observed a significant reduction in the conductance asymmetry of the trichome.

We believe two key sentences in the original manuscript were in part the source of the confusion. These sentences were modified as follow.

The sentence: “The resulting cavitation of the dome cell shifts the liquid-gas interface from the surface of the shield to the foot cell below (Fig. 2C), reducing the effective surface for evaporation to less than 3% of the total leaf surface area (Fig. 2D, Fig. S3). The geometrical advantage of reducing by 97% the evaporative surface highlights the importance of rectifying the flow of liquid water at the level of the dome cell for the trichome to act as a valve.” **was changed to:** (line 125) “The inward migration of the liquid-gas interface reduces the effective surface for evaporation to less than 3% of the total leaf surface area (Fig. 3d, Supplementary Fig. 4) and adds the thick shield walls as a new resistance for the outward diffusion of water vapor (Fig. 3a,c). These two effects explain the importance of rectifying the flow of liquid water at the level of the dome cell for the trichome to act as a valve.”

The sentence: “In summary, our results support the following mechanism for the conductance asymmetry: the main resistance to the entry of water is given by the plasma membrane of the foot cell (Fig. 2E and Supp. Text) [21], while the main resistance to evaporation comes from the extreme reduction of the evaporative surface by the cavitation of the dome cell and the accrued resistance of the shield wall at low relative humidity.” **was changed to:** (line 146) “ In summary, our results support the following mechanism for the conductance asymmetry: the main resistance to the entry of water is given by the plasma membrane of the foot cell (Fig. 3e and Supplementary Note)[Benzing1970], while the main resistance to evaporation comes from the reduction of the evaporative surface by the migration of the liquid-gas interface and the addition of the shield wall resistance to the diffusion path.”

Also, the simple biomimetic system proposed by the authors seems to be missing both key ingredients discussed above. It is not clear whether cavitation occurs in this system (is it possible to see the air space filling and emptying during operation?), and if I understand correctly, there is no difference in evaporation area between the cellulose layer and the semipermeable membrane.

The reviewer is correct that we did not implement explicitly an air space to mimic the function of the dome cell, nor was it possible to see the air within the composite membrane of the biomimetic system (the system is now shown in Supplementary Figure 7). The air space did not have to be implemented because of the large difference in scale between the trichome and our system. For comparison, the thickness of a single sheet of paper in the biomimetic system already exceeds the size of the flow path for the entire *Tillandsia* trichome. The main issue for the large scale biomimetic system was not to produce cavitation but in fact to remove air bubbles trapped between the sheets of paper or between the paper and the semipermeable membrane. From the basic physics of Laplace pressure, it is known that large bubbles (say 1mm in size) tend to be stable because of their low internal pressure. In contrast, a small bubble that might be

found within a trichome (say $1\ \mu\text{m}$ in size) will have a substantially higher Laplace pressure which forces gases within the bubble in solution, leading the bubble to collapse on itself. Therefore, cavitation or air-seeding at the scale of the dome cell is a much challenging problem than air-seeding in a macroscopic system.

In a properly functioning *Tillandsia* trichome, the surface over which water is absorbed and evaporated is the same; and corresponds to the membrane of the outermost foot cell. Since the walls of the trichome are cutinized, the movement of water must ultimately pass through the bottleneck of the foot cell whether in absorption or evaporation. Therefore, the conductance asymmetry measured in normally functioning trichomes is not imputable to a change in area. Under the “abnormal” wick scenario, the area of the evaporation surface would in fact be larger than the area of the absorption surface, and therefore detrimental to the conduction asymmetry. For the biomimetic system, the absorption and evaporation is done at the semipermeable membrane so the area is the same as for a properly functioning *Tillandsia* trichome. The main difference is that in the trichome the diffusion path opens up in the shield while in the biomimetic system, the diameter of the diffusion path remains the same. However, this difference is of minor consequences.

To demonstrate the role of the air space as a flow rectifier, the biomimetic experiments should also include a case where the air space is removed between the cellulose layers and the semipermeable membrane.

As stated above, removing the “air space” in the biomimetic system would be nearly impossible because of its large scale. The cellulose layers and semipermeable membrane would have to be produced and bound with micrometric precision. We do have, however, indirect evidence of the importance of air spaces. The semipermeable membrane used in the biomimetic system does not have a perfect reflection coefficient for NaCl. Therefore, a small amount of salt migrates from the reservoir into the composite membrane. When a critical amount of salt builds up in the membrane, it provides an osmotic potential that allows water to remain present between the cellulose layers and the semipermeable membrane. We have noted that when this happens, the rate of evaporation is increased substantially. In our experiments, we were careful to change our systems regularly to avoid salt buildup.

Another thing that was unclear is where the water absorbed during intake events is stored, both in real leaves and in the biomimetic system: is the osmotic solution diluted, and if so shouldn't the driving force decrease over time?

Yes, the reviewer is correct, but the effect is minor. In the case of the biomimetic system, the chamber contains a large volume of air allowing for the filling of liquid water. While the increase of the volume of water leads to an increase (less negative) of the osmotic potential of the chamber, this effect can be neglected because the increase in water mass is on the order of 45 mg per hour while the total volume of water is about 15 000 mg. In the case of the *Tillandsia* leaves, the water that enters the mesophyll can be stored in part in the so-called hygrenchyma below the adaxial face of the leaf. Although leaves can to some extent regulate their water

potential, it is clear that long term exposure to water leads to an increase (less negative) in the water potential. Note that the reduction in the osmotic driving force is clearly visible in Fig. 2a (formerly Fig. 1F) as a gradual change in the slope of leaf mass vs time. In this experiment, water absorption was performed during an extended period of time (10 h of cumulative exposure), causing a significant decrease in the absorption rate. For all other measurements, the water absorption was performed over a much shorter time in order to avoid, in particular, such changes in water potential.

In addition, it is clear that the leaves used in our experiments were in slightly distinct hydration states. Normalizing for such effect would have required to measure the leaf's water potential for every experiment. Such measurements are tedious. The work-around used by plant physiologists is to measure the dry weight (and sometimes the saturated weight) of the leaf after completion of the experiments (see for example Biebl, R. (1964). *Zum Wasserhaushalt von Tillandsia recurvata L. und Tillandsia usneoides L. auf Puerto Rico*. *Protoplasma* 58: 345-368.). We have followed this protocol and listed the dry weight of the leaf in what is now Fig. 2a where this information seemed useful. Elsewhere, the slight changes in the water potential of the leaves seem of little relevance so the dry weights were not listed.

Finally, the authors discuss potential applications for water harvesting in the conclusion. A variety of designs have been proposed in recent years for fog harvesting and it would be useful to compare the strategy elucidated here to these other strategies. An obvious downside seems to be that since osmosis is the main driving force, the harvested water contains solutes that have to be filtered out later.

Tillandsia does not compare advantageously with, for example, the passive mesh collectors used to capture fog droplets (e.g. Schemenauer R. S. and Joe, P. I., 1989. The collection efficiency of a massive fog collector. *Atmos. Res.* 24, 53–69). The liquid water flux of fog is about $100 \text{ g m}^{-2} \text{ min}^{-1}$. Since mesh collectors have an efficiency of 20%, they capture fog water at a rate of about $20 \text{ g m}^{-2} \text{ min}^{-1}$. In contrast, the measured absorption rate of fog water in *Tillandsia* is only $0.24 \text{ g m}^{-2} \text{ min}^{-1}$. Moreover, the water captured by mesh collectors can be directed to a sealed reservoir so the rate of water loss by evaporation is nearly zero. Therefore, it could be argued that even a passive fog collector has a far superior “transport” asymmetry than *Tillandsia*.

However, the *Tillandsia* plant is much more than a simple fog collector since the dense trichome layer allows the plant to exchange with its environment while maintaining an internal water potential well above the average water potential of the environment. We believe a fairer comparison is with other plants that face the same challenges of acquiring resources from their environment (water, minerals, CO_2 , etc) while also maintaining a relatively high water potential. We therefore added a comparison with perhaps the best studied water acquisition structure of desert plants, the roots of succulents. This comparison can be found in the discussion (lines 214-249).

I also have a few more technical questions or remarks:

- What is the basis for defining water conductance with respect to water potential? It seems to me

that this definition artificially amplifies the asymmetry between uptake and evaporation (the 5800 factor) due to the nonlinearity of water potential with RH.

It is true that the word “conductance” can refer to many different physical properties, each with its own units. We believe our definition, which includes the difference in potential, is the most common usage in plant physiology and physics.

For example, the Oxford Dictionary defines electrical conductance as:

“The degree to which an object conducts electricity, calculated as the ratio of the current which flows to the potential difference present.”

In “Plant Physiology” by Taiz and Zeiger, hydraulic conductivity is defined as: “Hydraulic conductivity expresses how readily water can move across a membrane and has units of volume of water per unit area of membrane per unit time per unit driving force (for instance, $\text{m}^3 \text{m}^{-2} \text{s}^{-1} \text{MPa}^{-1}$ or $\text{m s}^{-1} \text{MPa}^{-1}$).” (<http://6e.plantphys.net/topic03.09.html>). In our case, the word “conductance” is more appropriate than conductivity because the trichome is a heterogeneous structure with a least three distinct intrinsic conductivities.

Irrespective of how conductance is defined, the most fundamental question raised by the reviewer is how should the transport asymmetry of the trichome valve be defined. Our logic for using the ratio of the conductance is as follows. We are looking for a measure of asymmetry that is intrinsic to the trichome and depends the least possible on the conditions under which it is obtained. If, for example, the trichome asymmetry was defined just as the ratio of fluxes, its value would change depending on the RH used in the evaporation phase. For the conditions under which the trichome was tested (RH=50%), the gradient in potential driving evaporation is nearly 100 times greater than the gradient driving liquid water absorption. Thus, the intrinsic conductance asymmetry of the trichome is partly countered by the clear potential bias in favor of evaporation. Looking at it from another perspective, an asymmetry based on fluxes could be brought artificially in favor of absorption by making the measurement of evaporation near 100% RH. Under such conditions, the trichome would still absorb liquid water but the evaporative flux would be negligible. The asymmetry based on the classical definition of conductance alleviates this problem and, in our mind, is the only definition that does not “artificially amplifies the asymmetry”.

- Is there any reason why (and could you confirm that) freeze-thaw procedures only rupture plasma membranes and not other components?

We did not attempt to identify specifically what structures were disrupted by the freeze-thaw experiment since it would require the difficult task of preparing TEM images under precisely controlled experimental conditions. Rather, our interpretation is based on the extensive literature on frost damage in plants. Freezing induces many physiological stresses in living cells but it is generally agreed that dehydration of the cell content and a concomitant destabilization of the plasma membrane are among the major stressors that can ultimately lead to cell death (Steponkus, P. L. (1984). Role of the plasma membrane in freezing injury and cold acclimation.

Annual Review of Plant Physiology, 35(1), 543-584). We emphasize that frozen cells do not burst as would a glass bottle filled with water. On the contrary, the growth of ice crystals in the apoplast pulls water molecules from the cell sap hence leading to dehydration of the cell. In the case of our experiments, the leaves are frozen rapidly to a temperature of -80°C , well below the typical critical temperature for freezing of the cell content. We therefore expect all hydrated membrane-bound structures such as the plasma membrane, mitochondria and chloroplasts, to be disrupted. On the other hand, the waxy cuticle and the cellulosic wall of the shield cells contain little liquid water under our experimental conditions. Therefore, these structures should not be damaged during the freezing process.

- The expected effect of CuSO_4 treatment should be described more clearly. Why would it increase evaporation flux? Why would it form crystals as in Fig S5? And why the choice of this particular compound?

Cu^{2+} are large ions that cannot easily cross the cell membrane at the concentration used. Commonly used as a fertilizer or fungicide, CuSO_4 is bio-compatible and not expected to harm the plant. This is why this salt was chosen. The expected effect of the CuSO_4 treatment is that the accumulation of CuSO_4 in the lumen of the dome cell prevents the latter from drying normally. As a consequence, water with low chemical potential remains in the lumen during evaporation phase, which prevent the lumen from separating the cellulose from the semi-permeable membrane.

The following description was added to the main text:

“The salt is expected to travel freely across walls but Cu^{2+} should not cross membranes easily at this concentration [Lin2003]. Leaves were able to draw water from this weak CuSO_4 solution (Fig. 4c) but, as expected, Cu^{2+} accumulated within the lumen of the dome cell rather than cross the cell membrane (Supplementary Fig. 6). The gradual accumulation of ions allows the dome cell to develop a more negative osmotic potential, thus maintaining liquid water at this level. The presence of liquid water explains the slight increase in the rate of evaporative loss after each exposure to the solution, reaching evaporation rates characteristic of the freeze-thaw experiments (Fig. 4c,d).”

The legend of Fig. S5 (now Supplementary Fig. 6) was also corrected. We expect Cu^{2+} to accumulate at the first membrane if encounters but crystals are not easily seen in the picture of the trichome.

- How are the water sorption measurements made? It's also not obvious how these results help understanding the function of trichomes, as they are not used in the discussion except for illustrating that the cellulose layers are hygroscopic.

Trichomes were collected by shaving plants. After drying them in an oven, they were placed in a chamber with different preset relative humidities and equilibrated for at least 1 hour before recording their weight. The resulting sorption curve shows the characteristic increase of mass

commonly observed with cellulose. The sorption curve highlights the hygroscopic nature of the trichome shield, and allows us to develop a model based on the hypothesis that trichome shields behave as other cellulosic material. The sorption curve is necessary in particular to explain why the conductance of the leaf and of our biomimetic system is decreased at low relative humidity. This explanation can be found starting at line 131 of the main text.

- I did not understand the argument used by the authors against the “plug” scenario for the trichome function: overlapping of the trichome wings is not a problem if all the trichomes have a coordinated motion. Also, are the wings the only external extensions of the trichomes or is there a “hair” part not represented on the schematics?

Wings are the only extension of the trichomes. Figure 1d,e and Supplementary Fig. 2a,b represent the section of the complete trichome while the extent of the trichome from a top view can be seen in Supplementary Fig. 3 (trichomes on the left border). In some of our diagrams, the wing cells are not included to show with greater clarity the details of the central shield cells and dome cell.

Our argument against the old mechanism for the valve action is threefold. First, in *Tillandsia aeranthos* and *T. landbeckii*, the trichomes are so densely packed that their central shields and wings are often forced out of the plane of the epidermis. It is not clear how a good seal would be maintained with such distorted trichomes. Mez’s purported valve movement for the trichome does not compare well with a true plant valve such as the stomata where the lips of the guard cells are designed to close together with great precision and need to do so to avoid transpiration. Second, it is not clear why the trichome would need to “open” at all since, as our experiments show, the shield walls are highly permeable to water and do not contribute significantly to the total resistance of the flow path. Therefore the proposal that the trichome would have to “open” to let liquid water in the leaf is inconsistent with our measurements. Third, even if the movement of the trichomes were to contribute slightly to the asymmetry in the water conductance, we would still need to explain why the highly hygroscopic shield walls do not wick up water from the leaf’s mesophyll. Therefore, even under the best scenario, Mez’ mechanism is incomplete.

- The methods or supplementary material would benefit from addition of a picture of the biomimetic device. Also, figures S1-S5 lack information of scale, and it’s not obvious what we should be observing in the sequence in Figure S1B.

We added a picture of the biomimetic design as well as a cross-section of the composite membrane through which water must flow (Supplementary Fig. 7).

We added scale bars to all the figures that needed one.

Fig. S1B was added to make two points. First, unlike most trichomes and leaf surfaces, the trichomes on the *Tillandsia* leaf are highly hygroscopic and therefore allow fog droplets to form a thin film of water on the leaf surface. Second, in the process of drying, after fog exposure, the

thin film of free surface water must evaporate first before the true resistance of the leaf can be measured. This first phase is labeled Boundary Layer Resistance in the table of Fig. 4d.

- Why would the pathways for evaporation (e.g. Fig. S7) be different from the pathways for water absorption?

At the leaf level, the path for water evaporation has two features not shared with the absorption of liquid water. First, the evaporation path includes the resistance of the boundary layer above the leaf surface. This resistance is absent from the absorption path because liquid water arrives directly to the surface of the leaf. Second, the stomata are not thought to be able to absorb liquid water from the environment but they certainly offer an alternative path (parallel to the trichome path) for the evaporation of water. Despite the slightly different pathways between absorption and evaporation, our measurements and experimental manipulations indicate that the key elements for the transport asymmetry of the trichome are those illustrated in Fig. 3a.

- The authors mention concerns about the boundary layer outside of the leaf but then seem to exclude that contribution. Could you clarify whether the boundary layer is a concern or not and how it affected early measurements?

We added air circulation in our evaporation experiments to make sure that the local water vapor concentration is well controlled (this point is made starting on line 298 of the Methods) and to fix the thickness of the boundary layer. Under forced air circulation, the boundary layer resistance is approximately three orders of magnitude smaller than the resistance of the cellulose shield (see the rates of evaporation in Fig. 4d). Therefore, in an experimental set-up equipped with a fan, the boundary layer can be neglected. Our main concern with the original fan-less set-up was the slow equilibration of the system, especially thinking about the complex twigs of *Tillandsia landbeckii* which could develop large boundary layers (i.e. local microclimates) if air circulation is not forced.

- Is the swelling of the cellulose network due to water absorption similar to the Bangham effect for porous media?

We thank the reviewer for pointing us to this area of material science. We believe a parallel could be drawn between the initial swelling phase of cellulose by ADSORPTION of water and the Bangham effect. However, the swelling of cellulose by ABSORPTION when the volume fraction of water becomes high is probably best compared to the swelling of polymeric gels. The reviewer may be interested in reading the following review:

Gor, G. Y., Huber, P., & Bernstein, N. (2017). Adsorption-induced deformation of nanoporous materials—A review. *Applied Physics Reviews*, 4(1), 011303.

where adsorption actuated cellulose materials are discussed alongside more classical examples of the Bangham effect.

Reviewer #3:

In this work, the authors have argued that the asymmetric water transport found on *Tillandsia aeranthos* can be explained by the three distinct cell structures. Based on this understanding of the water transport under different conditions, the authors have further demonstrated that a synthetic unidirectional water transport valve structure in a larger length scale works. This work contains an interesting understanding of asymmetric water transport through *T. aeranthos* leaves but I am not sure whether the understanding is solid using the current set of experiments and information about them. I postpone my decision until the authors address the following additional comments and questions below. With the authors' revised manuscript, I will check the novelty, scientific approaches, and impacts of this work holistically.

1. I cannot find the temperature condition of experiments including Figures 1F and 1G. The authors should add the temperature conditions for all their experiments that are influenced by relative humidity. It would be also great if the authors can compare them with the actual temperature conditions in the Atacama desert of Chile.

All experiment were conducted at a temperature of 22°C +/- 2°C. This statement now appears in the Methods on line 295. The temperature is comparable to the average daytime temperature in winter in the Atacama desert. However, temperature conditions in the Atacama vary significantly during the year and between daytime and nighttime. Near Copiapó where we collected the plants, the typical daily amplitude in temperature is 15°C, the yearly maximum is approx. 30°C and yearly minimum approx. 5°C.

The effect of temperature on air saturation is predictable through thermodynamics properties, which is why we worked at a fixed temperature. Moreover, the effect on the water vapor potential is, for the most part, already included by definition in the relative humidity, which is one reason we carefully measured and used this parameter rather than, for instance, water vapor concentration, or partial pressure.

2. Figure 4A should show the dimension of the composite membrane system, including the spacing between the cellulose layers and the bottom membrane. Showing the actual experimental setup is usually the best way. Figure 4B should show the relative humidity and other experimental conditions.

We included Supplementary Fig. 7 where details of the biomimetic system are shown. The spacing between the cellulose layers and semipermeable membrane is variable but on the scale of 100µm. The relative humidity for Fig. 4B (now Fig. 5b) was added in the legend.

3. I am not sure whether the authors actually used *T. landbeckii* for their quantitative experiments because all the figures regarding quantitative information indicated only *T. aeranthos*. Please clarify this point.

The reviewer is correct, the main results of the paper were obtained using *T. aeranthos*. The main reason for this is that the small leaves of *T. landbeckii* make most experimental manipulations difficult and estimations of the surface area of the complex twig would be fairly imprecise. In addition, *T. landbeckii* was more difficult to maintain in a growing state in the laboratory. We have, however, performed water absorption and water evaporation experiments using *T. landbeckii* as well as structural analyses of the trichomes. These results are now shown in Supplementary Fig. 2. The micrographs and experimental results confirm that the structure and function of the trichomes of *T. landbeckii* and *T. aeranthos* are identical in all essential aspects.

We clarified the text by indicating in the figure legends of the main text what species was used. We also added this sentence to Fig. 1b “*Tillandsia aeranthos*, the plant studied most closely in this paper.” to avoid confusion. We would like to keep some mentions of *T. landbeckii* in the paper because it is the species that demonstrates most clearly that atmospheric bromeliads can rely almost exclusively on fog to fulfill their water needs. Other *Tillandsia* species such as *T. aeranthos* inhabit environments where rain is also a significant source of water.

4. One of the main factors regarding the evaporation suppression mechanism is the vacant space in the dome cell. If the authors’s explanation is correct, it would be possible to visualize this space with and without water using other types of visualization technique such as 3D confocal microscopy. This would clearly support the authors’ argument and significantly help readers understand the authors’ explanation.

We agree with the referee that such observations would be useful and we had in fact attempted to provide microscopic evidence of the liquid-gas interface moving into the lumen of the dome cell but those attempts were unsuccessful. New attempts were made for the revised manuscript. We implemented two protocols using transmitted light microscopy. In one series of experiments, we first soaked leaf segments for at least 30 minutes in water to fill the trichomes with liquid water. We then prepared relatively thick paradermal sections of the leaf and immersed those sections in three types of hyper-osmotic solutions: NaCl, sucrose, and glycerol. As water is drawn out of the section, we expected to observed bubbles forming in the trichome. Such bubbles were never observed. In a second series of experiments, we simply observed the movement of the water menisci in air-dried paradermal sections. Although the menisci are easily seen when water still fills most of the trichome shields, the optics become increasingly poor as more air gets trapped within and between the shields. Ultimately, the optics is too poor to convincingly capture the meniscus migrating to the lumen of the dome cell.

Note that in previous work, we have successfully used the two experimental approaches described above to visualize cavitation. See for example:

Noblin, X, N Rojas, J Westbrook, C Llorens, M Argentina, and J Dumais. 2012. The fern sporangium: a unique catapult. *Science* 335: 1322.

Llorens, C, M Argentina, N Rojas, J Westbrook, J Dumais, X Noblin. 2016. The fern cavitation catapult: mechanism and design principles. *Journal of the Royal Society Interface* 13: 20150930.

As illustrated in Fig. 8a of Llorens *et al.* 2016, the change in refractive index between water and air yields a good contrast in transmitted light microscopy. *Tillandsia*, however, is a far more challenging structure to image because cavitation/air-seeding takes place deep within the leaf tissue.

5. Figure 1A and S1-S5 should have scale bars. In particular, Figure S1B should show the time information on each image.

Scale bars and time information were added as indicated.

Reviewers' comments:

Reviewer #1 (Remarks to the Author):

Second review

The authors revised the manuscript carefully and considered almost all suggestions given by the reviewer. The manuscript in the present form is clear and conclusive in the presentation and interpretation of data.

Recommendation for the editor is to publish the manuscript as it is.

Reviewer #2 (Remarks to the Author):

I appreciate the significant improvements made to the manuscript following all reviewers' remarks and I find them globally convincing. It is a nice series of experiments to characterize water absorption and evaporation in Tillandsia plants. However, I still have some concerns that I think would need to be addressed.

One thing I still find particularly confusing is the role of the lumen. While the functions of the plasma membrane as an osmotic membrane and of the shield wall as a barrier to vapor diffusion are clearly demonstrated through the series of experiments performed, the case of the lumen is unclear. I agree that the lumen is likely to empty during evaporation, but 1) the paper does not demonstrate that it actually does, and 2) I'm not sure whether it is crucial that it is present. In fact, the "biomimetic" experiments are performed without an air space and still show the same behavior. To me, it seems that the key point is that the cellulose shield wall desaturate during evaporation, which will happen in any case because it is in contact with low humidity vapor on its external surface. Thus, except if the authors can provide clear experimental evidence that the lumen has an important role, I would soften the claims about it by 1) just stating that it is likely to empty (actually, I would think that since it's not lignified, it could just collapse upon drying, is this possible?), 2) remove its role as a rectifier in the schematics (to me, that role is achieved by the shield wall), especially since it is drawn in the schematic of the biomimetic device while it is actually not present.

Following the remarks above, I would argue that the main components of the system are the plasma membrane and the shield wall, acting respectively as a semi-permeable osmotic membrane and as a porous medium with high permeability to liquid water but low permeability to vapor. As thus, the biomimetic system proposed here is actually simply a composite membrane as can be found in pervaporation or osmosis applications, and a word on this in the paper would be nice. The simple fact that the system works by osmosis during filling and by vapor diffusion during emptying is not stated clearly in the paper. I would provide this explanation as early as the abstract (while softening the role of the air space there).

On a related note, I find the statement "Semipermeable membranes and cellulosic walls offer, on their own, no particular asymmetry to the transport of water" misleading. While this is somewhat true for the osmotic membrane, as demonstrated by the authors experimentally, this is not true for the cellulose wall, because the situation considered here is vapor transport (emptying) vs. liquid transport (filling), and it is precisely because of the strong asymmetry in transport between these two situations that the composite membrane works well as a rectifier.

I also have more specific remarks that I list below.

- I find the new paragraph on page 7 particularly convoluted, with lots of different information

without clear logical flow. Since this is an important discussion, it would be useful to clarify it. Some elements are also a bit vague (e.g. what part of the system are we talking about, and is it during filling or evaporation). And the physical processes should be mentioned more clearly (e.g. osmotic flow vs. evaporation etc.).

- There are contradictory statements in the paper and author's response about the role and importance of the reduction of the surface area of evaporation vs absorption surface. Please clarify and homogenize.

- I would assume that emptying the lumen and the cellulose wall at the beginning of evaporation takes some time, it would be good to have an estimate of this transient both for the real leaf and the biomimetic system.

- Is it possible to have more information about the composition of the plasma membranes, and why they act as perfect semipermeable membranes (e.g. do they contain aquaporins?). This would help understand the statement that plasma membranes are much more permeable than synthetic ones, which lacks a reference.

- I think it still makes sense to mention in the paper the interesting comparison with usual fog harvesting strategies made in the author's response. Also, since fog harvesting strategies seem much more efficient than Tillandsia's strategy, I'm not sure I understand the last sentence "Were this water transport asymmetry deployed on a large scale etc."

- Most figures in the Supplementary Material could be bigger, especially Fig. S7, which too small to distinguish its contents.

- How is the contact between the composite membrane and the tube made watertight?

- It would be useful to have an estimate of the effective thickness of the turbulent boundary layer in the air.

- Page 19, is it possible to have a reference about the composition of printer paper?

Finally, here are some technical remarks and typos that I found:

- "absorption capacity" in page 10 suggests sorption / capacitance behavior. I think you're referring to permeability instead.

- I got confused by the use of "as for" in two different places on page 11, because this typically means "in reference to". The same paragraph and other places in the paper also have conflicts between the use of past and present tenses.

- In the Supplementary Material, please define "CAM".

- SI units for mass are kg, not g.

- Is "internal store" standard nomenclature?

Reviewer #3 (Remarks to the Author):

I have carefully examined the revised manuscript and the authors' response to my comments, in addition to their answers to the other two reviewers' questions.

The only but important issue is that it is not clear whether the dome cell lumen should be filled

with air (as illustrated in Figs. 3a & c) to play a unique "valve" role that reduces the evaporation rate reported in the experimental results. I think that the dome cell lumen can reduce the evaporation rate even if it is partially wetted by water, more like "intermediate state" of the wick and valve scenarios, or with some small wetted areas inside the dome cell lumen possibly due to capillary condensation. In other words, I completely agree with the second reviewer ("it is not even clear if the lumen space plays that much of a role in the asymmetry of fluxes between intake and evaporation: diffusion of water vapor through the cellulose network could be the limiting step in transport whether there is an air space below or not.")

In relation to this point, the authors were not able to fully address my comment #4. In addition, Supplementary Fig. 7 (in response to my comment #2) also does not show "an apparent, separate air space like the dome cell lumen". I think that even a porous layer on top of a semipermeable membrane, without a completely empty space (that is illustrated in Fig. 5a), can reduce the evaporation rate as it can lower the diffusion coefficient value. It is more appropriate to regard the dome cell lumen as part of "a porous layer composed of central shield cells, wing cells, and dome cell", which can show a partial wetting state.

In this context, "space" between the membrane and cellulose layers in Fig. 5a should be removed or modified according to Supplementary Fig. 7 as it is misleading.

In summary, I think the authors' valve scenario illustrated in Figs. 3a & c and Fig. 5a is not clearly supported by the experimental results. Therefore, I recommend that the authors revise the manuscript accordingly.

Other than that, there are some typographical errors, such as "ce;;s" in the Supplementary Fig.2 caption.

Design of a Unidirectional Water Valve in *Tillandsia*

Pascal S. Raux, Simon Gravelle, Jacques Dumais

Response to Reviewers

We thank the reviewers again for their insightful comments. We have made all the changes requested, including the modifications regarding the role of the dome cell.

Reviewer #1 (Remarks to the Author):

Second review

The authors revised the manuscript carefully and considered almost all suggestions given by the reviewer. The manuscript in the present form is clear and conclusive in the presentation and interpretation of data.

Recommendation for the editor is to publish the manuscript as it is.

We thank the reviewer for the positive review.

Reviewer #2 (Remarks to the Author):

I appreciate the significant improvements made to the manuscript following all reviewers' remarks and I find them globally convincing. It is a nice series of experiments to characterize water absorption and evaporation in *Tillandsia* plants. However, I still have some concerns that I think would need to be addressed.

One thing I still find particularly confusing is the role of the lumen. While the functions of the plasma membrane as an osmotic membrane and of the shield wall as a barrier to vapor diffusion are clearly demonstrated through the series of experiments performed, the case of the lumen is unclear. I agree that the lumen is likely to empty during evaporation, but 1) the paper does not demonstrate that it actually does, and 2) I'm not sure whether it is crucial that it is present. In fact, the "biomimetic" experiments are performed without an air space and still show the same behavior. To me, it seems that the key point is that the cellulose shield wall desaturate during evaporation, which will happen in any case because it is in contact with low humidity vapor on its external surface. Thus, except if the authors can provide clear experimental evidence that the lumen has an important role, I would soften the claims about it by 1) just stating that it is likely to empty (actually, I would think that since it's not lignified, it could just collapse upon drying, is this possible?), 2) remove its role as a rectifier in the schematics (to me, that role is achieved by the shield wall), especially since it is drawn in the schematic of the biomimetic device while it is actually not present.

Following the remarks above, I would argue that the main components of the system are the plasma membrane and the shield wall, acting respectively as a semi-permeable osmotic membrane and as a porous medium with high permeability to liquid water but low permeability to vapor. As thus, the biomimetic system proposed here is actually simply a composite membrane as can be found in pervaporation or osmosis applications, and a word on this in the paper would be nice. The simple fact that the system works by osmosis during filling and by vapor diffusion during emptying is not stated clearly in the paper. I would provide this explanation as early as the abstract (while softening the role of the air space there).

On a related note, I find the statement “Semipermeable membranes and cellulosic walls offer, on their own, no particular asymmetry to the transport of water” misleading. While this is somewhat true for the osmotic membrane, as demonstrated by the authors experimentally, this is not true for the cellulose wall, because the situation considered here is vapor transport (emptying) vs. liquid transport (filling), and it is precisely because of the strong asymmetry in transport between these two situations that the composite membrane works well as a rectifier.

We thank the reviewer for carefully spelling out his or her interpretation of the process. We have made the changes suggested. Specifically,

- Regarding the comment: “Thus, except if the authors can provide clear experimental evidence that the lumen has an important role, I would soften the claims about it by 1) just stating that it is likely to empty”

The claim about the dome cell was changed to (line 111) “the liquid-gas interface must shift from the surface of the trichome shield to somewhere within the dome cell”. Also, in Fig. 3c (ii), we indicated graphically that the dome cell is not necessarily emptied by leaving the meniscus of water within the lower half of the dome cell.

“2) remove its role as a rectifier in the schematics (to me, that role is achieved by the shield wall), especially since it is drawn in the schematic of the biomimetic device while it is actually not present.”

The rectifier schematics was removed from Fig. 3a and Fig. 5a.

- “As thus, the biomimetic system proposed here is actually simply a composite membrane as can be found in pervaporation or osmosis applications, and a word on this in the paper would be nice”.

Without a doubt the *Tillandsia* trichome and our bio-inspired system are simply composite membranes and as such they can be compared to other composite membranes already in use for osmosis and pervaporation. However, the composite membranes we describe seem to differ from others in one very fundamental way: our composite membranes need to present a conductivity asymmetry for a single molecular species (water in our case). In contrast, the essential ingredient for osmosis and pervaporation is selectivity, that is, the ability to let one molecular species through while remaining impermeable to others. Asymmetry in the

conductivity is not an issue in these latter examples. Given this important difference, we feel it would be confusing to discuss composite membranes whose main purpose is selectivity not. asymmetry.

- “The simple fact that the system works by osmosis during filling and by vapor diffusion during emptying is not stated clearly in the paper. I would provide this explanation as early as the abstract (while softening the role of the air space there)”

We added the following sentence to the abstract: “While fog absorption is achieved by capillary flow of liquid water, reversal of the flow under dry external conditions shifts the liquid-gas interface deep into the trichome thus forcing water to move through the thick trichome wall in the vapor phase.” Also, in the main text, starting at line 147, we added this sentence: “In summary, our results support the following mechanism for the conductance asymmetry: the entry of water occurs in the liquid phase with the main resistance given by the plasma membrane of the foot cell (Fig. 3e and Supplementary Note) [15], while the exit of water takes place in the vapor phase with the main resistance to evaporation coming from the diffusion of water through the thick shield walls.”

- “I find the statement “Semipermeable membranes and cellulosic walls offer, on their own, no particular asymmetry to the transport of water” misleading.”

We would like to clarify our use of the word “symmetric” and “asymmetric”. In our mind, both the cell wall and the plasma membrane, on their own, are symmetric because a strict reversal of the conditions on either side of the structure would lead to the same flow. Therefore, while it is true that the cell wall has a differential conductivity to liquid water versus water vapor, this is not per se an asymmetric permeability; it is simply an indication that the conductivity of the material changes for these two distinct phases of water. Ultimately, we focused our claim of “asymmetric transport” on the entire trichome: cell wall, dome cell, and plasma membrane; and did not attribute asymmetry to any of these structures individually.

Regarding the question “actually, I would think that since it’s not lignified, it could just collapse upon drying, is this possible?”, we believe collapse of the dome cell to be unlikely given its thick convex walls and solid anchoring in the shield and epidermis. At the very least, our observations on paradermal sections did not provide any evidence of collapsing dome cells. However, as we stated in our previous response, the optics through this complex tissue is poor and therefore it has not been possible to quantify any changes in dome cell size. We should add that a collapsing dome cell would be very detrimental to the function of the trichome as a water valve. To collapse, the dome cell would have to remain filled with liquid water and experience a negative pressure so that the thick elastic walls are bent inwards. Moreover, collapsing of the dome cell would have the added consequence of increasing the osmotic concentration of the lumen (i.e. lowering the osmotic potential) because the dissolved solutes would now be confined to a smaller lumen volume. The net result would be a highly negative water potential in the lumen of the dome cell which could then draw water from the mesophyll of the leaf.

I also have more specific remarks that I list below.

- I find the new paragraph on page 7 particularly convoluted, with lots of different information without clear logical flow. Since this is an important discussion, it would be useful to clarify it. Some elements are also a bit vague (e.g. what part of the system are we talking about, and is it during filling or evaporation). And the physical processes should be mentioned more clearly (e.g. osmotic flow vs. evaporation etc.).

The paragraph (starting line 111) was rewritten with a different focus and to emphasize the basic principle of the asymmetry between bulk osmotic flow and water vapor diffusion. We hope the logic is easier to follow in the latest version.

- There are contradictory statements in the paper and author's response about the role and importance of the reduction of the surface area of evaporation vs absorption surface. Please clarify and homogenize.

To avoid confusion, we have removed the comment about the reduction in surface area in the evaporation phase by the migration of the liquid-gas interface from the outer trichome to the foot cell.

- I would assume that emptying the lumen and the cellulose wall at the beginning of evaporation takes some time, it would be good to have an estimate of this transient both for the real leaf and the biomimetic system.

The evaporation of the "free" water associated with the air spaces at the surface of the leaf and imbibed into the shield's walls does indeed take some time. The timescale can be observed graphically in the inset of Fig. 2a and in Fig.4c. The volume of water, including the free surface water, represents approximately 10mg for a leaf of 250mg (fresh weight). The time scale over which the surface water evaporates is on the order of 30 min. The shape of the curve suggests that much of the surface water is simply maintained by capillarity. This is shown by a linear decrease of leaf mass. The exact volume of water present in the lumen of the dome cell and within the cell walls at the beginning of the drying process is not easy to estimate but from inspection of leaf sections, we can state that it represents less than 5% of the 10mg water held by capillarity at the surface of the leaf. Once more tightly held water fractions begin evaporating, the curve begins to follow an exponential (Fig. 2a, inset).

- Is it possible to have more information about the composition of the plasma membranes, and why they act as perfect semipermeable membranes (e.g. do they contain aquaporins?). This would help understand the statement that plasma membranes are much more permeable than synthetic ones, which lacks a reference.

As far as we can tell, the composition of the trichomes' plasma membranes in *Tillandsia* plants was never studied specifically. There is, however, indirect evidence that aquaporins are present in the plasma membrane. For example, it was shown that mercury, which is known to inhibit the function of aquaporins, has a negative impact on the rate of water absorption by the trichomes. See for example:

Ohrui, T., Nobira, H., Sakata, Y., Taji, T., Yamamoto, C., Nishida, K., ... & Tanaka, S. (2007). Foliar trichome- and aquaporin-aided water uptake in a drought-resistant epiphyte *Tillandsia ionantha* Planchon. *Planta*, 227(1), 47-56.

However, we would like to note that the plasma membrane with or without aquaporins is permeable to water. Therefore, although aquaporins can provide a certain level of control over the flow of water in and out of the trichome, they cannot explain the 5800 fold asymmetry in water conductance nor the rapid switch in conductance observed in the transition between absorption to evaporation.

- I think it still makes sense to mention in the paper the interesting comparison with usual fog harvesting strategies made in the author's response. Also, since fog harvesting strategies seem much more efficient than *Tillandsia*'s strategy, I'm not sure I understand the last sentence "Were this water transport asymmetry deployed on a large scale etc."

The sentence mentioned by the reviewer was removed. Instead, we added the following sentences at the end of the discussion (line 241): "Finally, we note that the rate of fog water interception by *Tillandsia* plants is comparable to meshes designed to capture fog water [25] (Supplementary Fig. 10) and that those rates of interception far exceed the rate at which water can be absorbed internally by the plant. It is therefore likely that *Tillandsia* satisfies its own water needs with fog water and also those of other plants and animals present in the vicinity."

This statement clarifies the confusion produced by our earlier response. As such, *Tillandsia* plants are not bad fog collectors in the sense that they can intercept fog droplets at the same rate as the meshes typically used to collect fog water. However, the rate of absorption of water into the leaves and stems is far lower than the rate of water interception. Therefore, a substantial fraction of the intercepted fog water simply remains on the surface of the plant or drops to the ground.

- Most figures in the Supplementary Material could be bigger, especially Fig. S7, which too small to distinguish its contents.

We increased the size of all the figures in the Supplementary Material.

- How is the contact between the composite membrane and the tube made watertight?

We modified text in the methods section to include a statement about the glue used to adhere the composite membrane to the tube. The text is (line 341):

"The biomimetic system consisted of a tubular reservoir filled with ~15mL of a 1M NaCl solution. One end of the tube was hermetically sealed while a composite membrane of area $A = (3.8 \pm 0.3) \text{ cm}^2$ was glued at the other end of the tube using acetoxy silicone adhesive (Loctite SI 5398, Henkel, Germany)."

- It would be useful to have an estimate of the effective thickness of the turbulent boundary layer in the air.

Since the width of the *Tillandsia* leaves is rather small (10mm), we estimated the thickness of the boundary layer to be approximately 1mm. The value appears in the section “Water evaporation.” of the Supplementary Material section. However, the thickness of the boundary layer is affected by the wind speed and wind direction so that only a range of boundary thickness can be confidently quoted. For example, Schuepp (1993) gives a range of 0.28-2.8mm for leaves of intermediate size (Schuepp, P. H., 1993. Tansley review No. 59. Leaf boundary layers. *New Phytologist*, 477-507.)

- Page 19, is it possible to have a reference about the composition of printer paper?

We added: Roberts, J. C. (2007). *The chemistry of paper*. Royal Society of Chemistry.

Finally, here are some technical remarks and typos that I found:

- “absorption capacity” in page 10 suggests sorption / capacitance behavior. I think you’re referring to permeability instead.

The words “absorption capacity” were replaced by “absorption rate” since this is what is reported in Fig. 4d.

- I got confused by the use of “as for” in two different places on page 11, because this typically means “in reference to”. The same paragraph and other places in the paper also have conflicts between the use of past and present tenses.

The words “as for” were replaced by “As observed for”. We also corrected the use of the past tense.

- In the Supplementary Material, please define “CAM”.

CAM stands for Crassulacean Acid Metabolism. The definition was added to the text following the abbreviation CAM.

- SI units for mass are kg, not g.

We are unsure what the reviewer is asking. Should we change all mentions of g and mg to kg?

- Is “internal store” standard nomenclature?

We replaced “internal stores” by mesophyll to be more specific.

Reviewer #3 (Remarks to the Author):

I have carefully examined the revised manuscript and the authors' response to my comments, in addition to their answers to the other two reviewers' questions.

The only but important issue is that it is not clear whether the dome cell lumen should be filled with air (as illustrated in Figs. 3a & c) to play a unique “valve” role that reduces the evaporation rate reported in the experimental results. I think that the dome cell lumen can reduce the evaporation rate even if it is partially wetted by water, more like “intermediate state” of the wick and valve scenarios, or with some small wetted areas inside the dome cell lumen possibly due to capillary condensation. In other words, I completely agree with the second reviewer (“it is not even clear if the lumen space plays that much of a role in the asymmetry of fluxes between intake and evaporation: diffusion of water vapor through the cellulose network could be the limiting step in transport whether there is an air space below or not.”)

In relation to this point, the authors were not able to fully address my comment #4. In addition, Supplementary Fig. 7 (in response to my comment #2) also does not show “an apparent, separate air space like the dome cell lumen”. I think that even a porous layer on top of a semipermeable membrane, without a completely empty space (that is illustrated in Fig. 5a), can reduce the evaporation rate as it can lower the diffusion coefficient value. It is more appropriate to regard the dome cell lumen as part of “a porous layer composed of central shield cells, wing cells, and dome cell”, which can show a partial wetting state.

In this context, “space” between the membrane and cellulose layers in Fig. 5a should be removed or modified according to Supplementary Fig. 7 as it is misleading.

In summary, I think the authors’ valve scenario illustrated in Figs. 3a & c and Fig. 5a is not clearly supported by the experimental results. Therefore, I recommend that the authors revise the manuscript accordingly.

We thank the reviewer for the useful comments. Figure 3c were modified to reflect the reviewer’s comments about the potential “small wetted areas with the dome cell lumen”. The air space in Fig. 5a was also removed. Finally, we modified the text to focus on the motion of the liquid-gas interface below the thick shield wall without claiming that the dome cell plays a direct role in the movement of the interface.

Other than that, there are some typographical errors, such as “ce;;s” in the Supplementary Fig.2 caption.

We revised the text carefully to remove the remaining typographical errors.

REVIEWERS' COMMENTS:

Reviewer #2 (Remarks to the Author):

The authors' response and revisions to the manuscript are satisfactory to me. I only regret that osmosis is not mentioned in the abstract as a driving force for water absorption. I would assume that « capillary flow » as mentioned in the abstract is only occurring for a short period of time (in a wicking/imbibition fashion) before a steady-state osmotic flow is reached. If I'm not mistaken, the authors have mostly characterized this osmotic flow, not capillary absorption.

Reviewer #3 (Remarks to the Author):

The authors have addressed all of my comments properly. I recommend this manuscript for publication.

Design of a Unidirectional Water Valve in *Tillandsia*

Pascal S. Raux, Simon Gravelle, Jacques Dumais

Response to reviewers

Reviewer #2 (Remarks to the Author):

The authors' response and revisions to the manuscript are satisfactory to me. I only regret that osmosis is not mentioned in the abstract as a driving force for water absorption. I would assume that « capillary flow » as mentioned in the abstract is only occurring for a short period of time (in a wicking/imbibition fashion) before a steady-state osmotic flow is reached. If I'm not mistaken, the authors have mostly characterized this osmotic flow, not capillary absorption.

We thank the reviewer for the pertinent comment. The summary now mentions osmosis:

“Here, we explain how a 5800-fold asymmetry in water conductance arises from a clever juxtaposition of a thick hygroscopic wall and a semipermeable membrane. While absorption is achieved by **osmosis** of liquid water, evaporation under dry external conditions shifts the liquid-gas interface forcing water to diffuse through the thick trichome wall in the vapor phase. ”

Reviewer #3 (Remarks to the Author):

The authors have addressed all of my comments properly. I recommend this manuscript for publication. Sincerely yours,

We thank the reviewer for his/her contribution to this manuscript